# Planning to the Information Horizon of BAMDPs via Epistemic State Abstraction

**Dilip Arumugam**
Department of Computer Science
Stanford University
dilip@cs.stanford.edu

**Satinder Singh**
DeepMind, London
baveja@deepmind.com

## Abstract

The Bayes-Adaptive Markov Decision Process (BAMDP) formalism pursues the Bayes-optimal solution to the exploration-exploitation trade-off in reinforcement learning. As the computation of exact solutions to Bayesian reinforcement-learning problems is intractable, much of the literature has focused on developing suitable approximation algorithms. In this work, before diving into algorithm design, we first define, under mild structural assumptions, a complexity measure for BAMDP planning. As efficient exploration in BAMDPs hinges upon the judicious acquisition of information, our complexity measure highlights the worst-case difficulty of gathering information and exhausting epistemic uncertainty. To illustrate its significance, we establish a computationally-intractable, exact planning algorithm that takes advantage of this measure to show more efficient planning. We then conclude by introducing a specific form of state abstraction with the potential to reduce BAMDP complexity and gives rise to a computationally-tractable, approximate planning algorithm.

## 1 Introduction

The Bayes-Adaptive Markov Decision Process (BAMDP) [Duff, 2002] is a classic formalism encapsulating the optimal treatment of the exploration-exploitation trade-off by a reinforcement-learning agent with respect to prior beliefs over an uncertain environment. Unfortunately, the standard formulation suffers from an intractably-large hyperstate space (that is, the joint collection of environment states coupled with the agent's current state of knowledge over the unknown environment) and much of the literature has been dedicated to identifying suitable approximations [Bellman and Kalaba, 1959, Dayan and Sejnowski, 1996, Duff and Barto, 1997, Dearden et al., 1998, Strens, 2000, Duff, 2001, 2003b,a, Wang et al., 2005, Poupart et al., 2006, Castro and Precup, 2007, Kolter and Ng, 2009, Asmuth et al., 2009, Dimitrakakis, 2009, Sorg et al., 2010, Araya-López et al., 2012, Guez et al., 2012, 2013, 2014, Ghavamzadeh et al., 2015, Zintgraf et al., 2019]. In this work, we take steps toward clarifying the hardness of BAMDPs before outlining an algorithmic concept that may help mitigate problem difficulty and facilitate near-optimal solutions.

First, we introduce the notion of *information horizon* as a complexity measure on BAMDP planning, characterizing when it is truly difficult to identify the underlying uncertain environment. Naturally, the agent's state of knowledge at each timestep (a component of the overall BAMDP hyperstate) reflects its current epistemic uncertainty and, as the agent accumulates data, this posterior concentrates, exhausting uncertainty and identifying the true environment; after this point, the Bayes-optimal policy naturally coincides with the optimal policy of the underlying Markov Decision Process (MDP). Simply put, the information horizon quantifies the worst-case number of timesteps needed for the agent to reach this point whereupon there is no more information to be gathered about the uncertain environment.

36th Conference on Neural Information Processing Systems (NeurIPS 2022).

With this complexity measure in hand, we then entertain the idea of *epistemic state abstraction* as an effective algorithmic tool for trading off between reduced information horizon (complexity) and near-Bayes-optimality of the corresponding planning solution. Intuitively, as the total number of knowledge states an agent may take on drives the intractable size of the hyperstate space, we operationalize state abstraction [Li et al., 2006, Abel et al., 2016] to perform a lossy compression of the epistemic state space, inducing a "smaller" and more tractable BAMDP for planning; our results not only mirror those of analogous work on state aggregation for improved efficiency in traditional MDP planning [Van Roy, 2006] but also parallel similar findings [Hsu et al., 2007, Zhang et al., 2012] on the effectiveness of belief state aggregation in partially-observable MDP (POMDP) planning [Kaelbling et al., 1998].

On the whole, our work provides one possible answer to a question that has already been asked and answered several times in the context of MDPs [Bartlett and Tewari, 2009, Jaksch et al., 2010, Farahmand, 2011, Maillard et al., 2014, Bellemare et al., 2016, Arumugam et al., 2021, Abel et al., 2021]: how hard is my BAMDP? While the remainder of the paper goes on to examine how one particular mechanism for reducing this complexity can translate into a more efficient planning algorithm, we anticipate that this work can serve as a starting point for building a broader taxonomy of BAMDPs, paralleling existing structural classes of MDPs [Jiang et al., 2017, Sun et al., 2019, Agarwal et al., 2020, Jin et al., 2021].

## 2 Problem Formulation

In this section, we formally define BAMDPs as studied in this paper. As a point of contrast, we begin by presenting the standard MDP formalism used throughout the reinforcement-learning literature [Sutton and Barto, 1998]. We use $\Delta(\mathcal{X})$ to denote the set of all probability distributions with support on an arbitrary set $\mathcal{X}$ and denote, for any natural number $N \in \mathbb{N}$, the index set as $[N] = \{1, 2, \ldots, N\}$. For any two arbitrary sets $\mathcal{X}$ and $\mathcal{Y}$, we denote the class of all functions mapping from $\mathcal{X}$ to $\mathcal{Y}$ as $\{\mathcal{X} \to \mathcal{Y}\} \triangleq \{f \mid f : \mathcal{X} \to \mathcal{Y}\}$.

### 2.1 Markov Decision Processes

We begin with a sequential decision-making problem represented via the traditional finite-horizon Markov Decision Process (MDP) [Bellman, 1957, Puterman, 1994] $\langle \mathcal{S}, \mathcal{A}, \mathcal{R}, \mathcal{T}, \beta, H \rangle$ where $\mathcal{S}$ is a finite set of states, $\mathcal{A}$ is a finite set of actions, $\mathcal{R} : \mathcal{S} \times \mathcal{A} \to [0, 1]$ is a deterministic reward function, $\mathcal{T} : \mathcal{S} \times \mathcal{A} \to \Delta(\mathcal{S})$ is a transition function prescribing next-state transition distributions for all state-action pairs, $\beta \in \Delta(\mathcal{S})$ is an initial state distribution, and $H \in \mathbb{N}$ is the horizon denoting the agent's total number of steps or interactions with the environment. An agent's sequential interaction within this environment proceeds in each timestep $h \in [H]$, starting with an initial state $s_1 \sim \beta(\cdot)$, by observing the current state $s_h \in \mathcal{S}$, selecting an action $a_h \in \mathcal{A}$, and then enjoying a reward $\mathcal{R}(s_h, a_h)$ as the environment transitions to $s_{h+1} \sim \mathcal{T}(\cdot \mid s_h, a_h)$. Action selections made by the agent are governed by its non-stationary policy $\pi$: a collection of $H$ stationary, deterministic policies $\pi = (\pi_1, \pi_2, \ldots, \pi_H)$, where $\forall h \in [H], \pi_h : \mathcal{S} \to \mathcal{A}$. We quantify the performance of policy $\pi$ at timestep $h \in [H]$ by its induced value function $V_h^\pi : \mathcal{S} \to \mathbb{R}$ denoting the expected sum of future rewards by deploying policy $\pi$ from a particular state $s \in \mathcal{S}$: $V_h^\pi(s) = \mathbb{E}\left[ \sum_{h'=h}^{H} \mathcal{R}(s_{h'}, a_{h'}) \mid s_h = s \right]$, where the expectation integrates over randomness in the environment transitions. Analogously, we define the action-value function induced by policy $\pi$ at timestep $h$ as $Q_h^\pi : \mathcal{S} \times \mathcal{A} \to \mathbb{R}$ which denotes the expected future sum of rewards by being in a particular state $s \in \mathcal{S}$, executing a particular action $a \in \mathcal{A}$, and then following policy $\pi$ thereafter: $Q_h^\pi(s, a) = \mathbb{E}\left[ \sum_{h'=h}^{H} \mathcal{R}(s_{h'}, a_{h'}) \mid s_h = s, a_h = a \right]$. We are guaranteed the existence of an optimal policy $\pi^\star$ that achieves supremal value $V_h^\star(s) = \sup_{\pi \in \Pi^H} V_h^\pi(s)$ for all $s \in \mathcal{S}, h \in [H]$ where the policy class contains all deterministic policies $\Pi = \{\pi \mid \pi : \mathcal{S} \to \mathcal{A}\}$. Since rewards are bounded in $[0, 1]$, we have that $0 \leq V_h^\pi(s) \leq V_h^\star(s) \leq H - h + 1$ for all $s \in \mathcal{S}, h \in [H]$, and $\pi$. These value functions obey the Bellman equation and the Bellman optimality equation, respectively:

$$V_h^\pi(s) = Q_h^\pi(s, \pi_h(s)) \quad V_h^\star(s) = \max_{a \in \mathcal{A}} Q_h^\star(s, a), \qquad V_{H+1}^\pi(s) = 0 \quad V_{H+1}^\star(s) = 0 \quad \forall s \in \mathcal{S},$$

$$Q_h^\pi(s,a) = \mathcal{R}(s,a) + \mathbb{E}_{s' \sim \mathcal{T}(\cdot|s,a)} \left[ V_{h+1}^\pi(s') \right], \qquad Q_h^\star(s,a) = \mathcal{R}(s,a) + \mathbb{E}_{s' \sim \mathcal{T}(\cdot|s,a)} \left[ V_{h+1}^\star(s') \right].$$

## 2.2 Bayes-Adaptive Markov Decision Processes

The BAMDP formalism offers a Bayesian treatment of an agent interacting with an uncertain MDP. More specifically, a decision-making agent is faced with a MDP $\mathcal{M} = \langle \mathcal{S}, \mathcal{A}, \mathcal{R}, \mathcal{T}_\theta, \beta, H \rangle$ defined around an unknown transition function $\mathcal{T}_\theta$[1], for some latent parameter $\theta \in \Theta$. Prior uncertainty in $\theta$ is reflected by the distribution $p(\theta)$. In classic work on BAMDPs with finite state-action spaces, the parameters $\theta$ denote visitation counts and $p(\theta)$ is a Dirichlet distribution, so as to leverage the convenience of Dirichlet-multinomial conjugacy for exact posterior updates [Duff, 2002, Poupart et al., 2006]. For our purposes, we will assume an alternative parameterization whose importance will be made clear later when defining our complexity measure.

**Assumption 1.** *We assume that $\Theta$ is known and $|\Theta| < \infty$ such that an agent is only ever reasoning about its uncertainty over a finite set of $|\Theta|$ known MDPs. We further make a realizability assumption that the true parameters reside in this finite set, $\theta \in \Theta$.*

Under Assumption 1, an agent's prior uncertainty in $\mathcal{T}_\theta$ is reflected by the distribution $p(\theta) \in \Delta(\Theta)$ which, with each step of experience encountered by the agent, may be updated via Bayes' rule to recover a corresponding posterior distribution in light of observed data from the environment. For simplicity, we do not concern ourselves with the computation of the posterior and instead assume access to a deterministic $\mathcal{B} : \Delta(\Theta) \times \mathcal{S} \times \mathcal{A} \times \mathcal{S} \to \Delta(\Theta)$ that performs an exact posterior update to any input distribution $p \in \Delta(\Theta)$ based on the experience tuple $(s,a,s') \in \mathcal{S} \times \mathcal{A} \times \mathcal{S}$ in $\mathcal{O}(1)$ time.

The corresponding BAMDP for $\mathcal{M}$ is defined around a so-called *hyperstate* space $\mathcal{X} = \mathcal{S} \times \Delta(\Theta)$ such that any hyperstate $x = \langle s, p \rangle \in \mathcal{X}$ denotes the agent's original or physical state $s \in \mathcal{S}$ within the true MDP while $p \in \Delta(\Theta)$ denotes the agent's information state or epistemic state [Lu et al., 2021] about the uncertain environment; intuitively, the epistemic state represents the agent's knowledge of the environment based on all previously observed data. This gives rise to the BAMDP $\langle \mathcal{X}, \mathcal{A}, \overline{\mathcal{R}}, \overline{\mathcal{T}}, \overline{\beta}, H \rangle$ where $\mathcal{A}$ is the same action set as the original MDP $\mathcal{M}$, $\overline{\mathcal{R}} : \mathcal{X} \times \mathcal{A} \to [0,1]$ is the same reward function as in $\mathcal{M}$ (that is, $\overline{\mathcal{R}}(\langle s, p \rangle, a) = \mathcal{R}(s,a) \; \forall \langle s, p \rangle \in \mathcal{X}, a \in \mathcal{A}$), $\overline{\beta} \in \Delta(\mathcal{X})$ is defined as $\overline{\beta} = \beta \times \delta_{p(\theta)}$ where $\delta_{p(\theta)}$ denotes a Dirac delta centered around the agent's prior $p(\theta)$, and $H$ is the same horizon as MDP $\mathcal{M}$. Due to the determinism of the posterior updates given by $\mathcal{B}$, the BAMDP transition function $\overline{\mathcal{T}} : \mathcal{X} \times \mathcal{A} \to \Delta(\mathcal{X})$ is defined as

$$\overline{\mathcal{T}}(x' \mid x, a) = \sum_{\theta \in \Theta} \mathcal{T}_\theta(s' \mid s, a) p(\theta) \mathbb{1}\left( p' = \mathcal{B}(p, s, a, s') \right),$$

where $x' = \langle s', p' \rangle \in \mathcal{X}$. The associated BAMDP policy $\pi = (\pi_1, \pi_2, \ldots, \pi_H)$, $\pi_h : \mathcal{X} \to \mathcal{A}, \forall h \in [H]$ selects actions based on the current state of the MDP as well as the agents accumulated knowledge of the environment thus far. With these components, we may define the associated BAMDP value functions with $x' = \langle s', \mathcal{B}(p, s, a, s') \rangle$:

$$V_h^\pi(x) = Q_h^\pi(x, \pi_h(x)) \quad V_h^\star(x) = \max_{a \in \mathcal{A}} Q_h^\star(x, a), \qquad V_{H+1}^\pi(x) = 0 \quad V_{H+1}^\star(x) = 0 \quad \forall x \in \mathcal{X},$$

$$Q_h^\pi(x, a) = \mathcal{R}(s,a) + \sum_{s', \theta} \mathcal{T}_\theta(s' \mid s, a) p(\theta) V_{h+1}^\pi(x'), \qquad Q_h^\star(x, a) = \mathcal{R}(s,a) + \sum_{s', \theta} \mathcal{T}_\theta(s' \mid s, a) p(\theta) V_{h+1}^\star(x').$$

Based on these optimality equations, we see that the optimal policy of a BAMDP achieving supremal value $V_h^\star$ across all timesteps $h \in [H]$ is the Bayes-optimal policy which appropriately balances the exploration-exploitation trade-off in reinforcement learning. An observation is that this Bayes-optimal policy will tend to achieve lower value than the optimal policy of MDP $\mathcal{M}$ as the agent takes more informative (possibly sub-optimal) actions to identify the true underlying environment.

# 3 Related Work

Bellman and Kalaba [1959] offer the earliest formulation of Bayesian reinforcement learning, whereby the individual actions of a decision-making agent not only provide an update to the physical state

---

[1]For ease of exposition, we focus on uncertainty in transition dynamics although one could model uncertainty in either the reward function or the full MDP model (rewards & transitions).

of the world but also impact the agent's internal model of how the world operates. Dayan and Sejnowski [1996] follow this line of thinking to derive implicit exploration bonuses based on how an agent performs posterior updates. Kolter and Ng [2009] make this more explicit and incorporate a specific visitation-based bonus that decays with the concentration of the agent's Dirichlet posterior. As an alternative, Sorg et al. [2010] incorporate an exploration bonus based on the variance of the agent's posterior while Araya-López et al. [2012] achieve optimistic exploration by boosting transition probabilities. Duff and Barto [1997] identify multi-armed bandits (that is, MDPs with exactly one state and arbitrarily many actions) as a unique setting where the Bayes-optimal solution is computationally tractable through the use of Gittins indices [Gittins, 1979]. While the vast space of more complicated BAMDPs are computationally intractable, a goal of this paper is to add a bit of nuance and clarify when one might still hope to recover efficient, approximate planning. This is also distinct from the PAC-BAMDP framework introduced by Kolter and Ng [2009], which serves as a characterization of algorithmic efficiency, rather than problem hardness.

Representing uncertainty in the optimal value function rather than environment transition function, Dearden et al. [1998] derive a practical Bayesian $Q$-learning algorithm by foregoing representation of the epistemic state and instead resampling $Q^\star$-values at each timestep. Strens [2000] finds an alternate, tractable solution by lazily updating the epistemic state at the frequency of whole episodes, rather than individual timesteps; a long line of work [Agrawal and Jia, 2017, Osband et al., 2016a,b, Osband and Van Roy, 2017, O'Donoghue et al., 2018, Osband et al., 2019] analyzes this type of approximation to the Bayesian reinforcement-learning problem theoretically and also explores how to scale these solution concepts with deep neural networks.

Duff [2001] finds tractability in representing policies as finite-state stochastic automata, noting structural similarities between BAMDPs and partially-observable MDPs (POMDPs) [Kaelbling et al., 1998]; this type of thinking is further extended by Poupart et al. [2006] who exploit similar structure between the optimal value functions of BAMDPs and POMDPs. Duff [2003a] examine improved memory requirements when applying actor-critic algorithms [Konda and Tsitsiklis, 2000] to BAMDPs while Duff [2003b] consider how to approximately model the stochastic process of the evolving epistemic state via diffusion models. Wang et al. [2005] introduce a sparse-sampling approach [Kearns et al., 2002] for balancing computational efficiency against fidelity to Bayes-optimal action selection. An analogous sparse-sampling approach is also developed by Castro and Precup [2007], but with a linear-programming methodology for value-function approximation. A line of work [Guez et al., 2012, 2013, 2014] develops more scalable, sparse-sampling lookahead approaches on the back of Monte-Carlo tree search [Kocsis and Szepesvári, 2006]; these algorithms are somewhat similar in spirit to the approach of Asmuth et al. [2009] who merge multiple posterior samples into a single model while Guez et al. [2014] keep each sample distinct and integrate out the posterior randomness. For a more complete and detailed survey of Bayesian reinforcement learning, we refer readers to Ghavamzadeh et al. [2015]. Crucially, the aforementioned approaches largely revolve around ignoring the epistemic state, lazily updating the epistemic state, or approximating the impact of the epistemic state via random sampling. In contrast, this work offers a new approach and highlights how lossy compression of the epistemic state may naturally reduce BAMDP hardness. Perhaps the most related prior work is by Lee et al. [2018] who introduce a practical approximate-planning approach by quantizing the epistemic state space; this paper clarifies the theoretical ramifications of this quantization step.

Our work is also connected to analyses of approximate value iteration [Bellman, 1957] in the MDP setting [Tseng, 1990, Littman et al., 1995], where more recent work has managed to recover improved sample complexity bounds for approximate value iteration [Sidford et al., 2018b,a]. Like Kearns and Singh [1999], our algorithms utilize exact value iteration almost as a black box and it is an open question for future work to see if similar ideas and proof techniques for these approximate variants might be leveraged in the BAMDP setting. Crucially, the variants of value iteration introduced in this work are merely a backdrop for illustrating the utility of our complexity measure and, more generally, a regard for underlying information structure in BAMDPs.

The particular class of epistemic state abstraction introduced and studied in this work revolves around the covering number of the epistemic state space. Curiously, this deepens an existing connection between BAMDPs and POMDPs [Duff, 2001], where a line of work establishes the covering number of the belief state space as a viable complexity measure for the latter both in theory [Hsu et al., 2007] and in practice [Zhang et al., 2012]. In a less related but similar vein, Kakade et al. [2003a] establish a provably-efficient reinforcement-learning algorithm when the MDP state space is a metric space;

their corresponding sample complexity guarantee depends on the covering number of the state space under the associated metric. Our planning complexity result for abstract BAMDPs mirrors those established by these works in its dependence on the covering number of the epistemic state space.

# 4 The Complexity of BAMDP Planning

In this section, we examine the difficulty of solving BAMDPs through the lens of a classic planning algorithm: value iteration [Bellman, 1957]. Due to space constraints, we relegate pseudocode for all discussed algorithms to Appendix A. We begin with an quick review of the traditional algorithm applied to our setting before introducing the information horizon as a complexity measure for BAMDPs. This quantity gives rise to a more efficient planning algorithm for BAMDPs that waives excessive dependence on the original problem horizon. In order to facilitate an analysis of planning complexity in BAMDPs via value iteration, we require a finite hyperstate space $\mathcal{X}$. For now, we will assume that $\mathcal{X}$ is finite, but still considerably large, by virtue of an aggressively-fine quantization of the $(|\Theta| - 1)$-dimensional simplex, also considered in the empirical work of Lee et al. [2018]:

**Assumption 2.** *We assume the existence of a suitable, fixed quantization of simplex $\widehat{\Delta}(\Theta) \subset \Delta(\Theta)$ where $|\widehat{\Delta}(\Theta)| < \infty$ such that the BAMDP hyperstate space $\mathcal{X} = \mathcal{S} \times \widehat{\Delta}(\Theta)$ is finite, $|\mathcal{X}| < \infty$.*

## 4.1 Naive Value Iteration

To help build intuitions, we begin by presenting a typical version of value iteration for finite-horizon BAMDPs as Algorithm 1. This algorithm iterates backwards through the $H$ timesteps, computing $Q_h^\star$ across every hyperstate-action pair. With the provision of our posterior update oracle $\mathcal{B}$, we avoid a square dependence on the hyperstate space $(|\mathcal{X}|^2)$ and instead only require $\mathcal{O}(|\mathcal{S}||\Theta|)$ to compute next-state value. Consequently, the resulting planning complexity of Algorithm 1 is $\mathcal{O}(|\mathcal{X}||\mathcal{A}||\mathcal{S}||\Theta|H)$. Clearly, this represents an onerous burden for two distinct reasons: (1) we are forced to contend with a potentially very large horizon $H$ and (2) we must also search through the entirety of the hyperstate space, $\mathcal{X}$. In the sections that follow, we alleviate the burdens of challenges (1) and (2) in series, using our new notion of information horizon to mitigate the impact of $H$ and leveraging epistemic state abstraction to further reduce the role of $|\mathcal{X}|$, where the latter occurs at the cost of introducing approximation error.

## 4.2 Information Horizon

As noted in the previous section, our planning complexity suffers from its dependence on the BAMDP horizon $H$. A key observation, however, is that once an agent has completely resolved its uncertainty and identified one of the $|\Theta|$ environments, all that remains is to deploy the optimal policy for that particular MDP. As an exaggerated but illustrative example of this, consider a BAMDP where any action executed at the first timestep completely identifies the true environment $\theta \in \Theta$. With no residual epistemic uncertainty left, the Bayes-optimal policy would now completely coincide with the optimal policy and take actions without changing the epistemic state since, at this point, the agent has acquired all the requisite information about the previously unknown environment. Even if the problem horizon $H$ is substantially large, a simple BAMDP like the one described should be fairly easy to solve as epistemic uncertainty is so easily diminished and information is quickly exhausted; it is this principle that underlies our hardness measure.

Let $\pi$ be an arbitrary non-stationary policy. For any hyperstate $x \in \mathcal{X}$, we denote by $\mathbb{P}^\pi(x_h = x)$ the probability that policy $\pi$ visits hyperstate $x$ at timestep $h$. With this, we may define the reachable hyperstate space of policy $\pi$ at timestep $h \in [H]$ as $\mathcal{X}_h^\pi = \{x \in \mathcal{X} \mid \mathbb{P}^\pi(x_h = x) > 0\} \subset \mathcal{X}$. In words, the reachable hyperstate space of a policy $\pi$ at a particular timestep is simply the set of all possible hyperstates that may be reached by $\pi$ at that timestep with non-zero probability. Recall that for any hyperstate $x = \langle s, p \rangle \in \mathcal{X}$, the epistemic state $p \in \Delta(\Theta)$ is a (discrete) probability distribution, for which we may denote its corresponding entropy as $\mathbb{H}(p)$. Given a BAMDP, we define the *information horizon of a policy* $\pi$ as $\mathcal{I}(\pi) = \inf\{h \in [H] \mid \forall x_h = \langle s_h, p_h \rangle \in \mathcal{X}_h^\pi, \mathbb{H}(p_h) = 0\}$. The information horizon of a policy, if it exists, identifies the first timestep in $[H]$ where, regardless of precisely which hyperstate is reached by following $\pi$ at this timestep, the agent has fully resolved all of its epistemic uncertainty over the environment $\theta$. At this point, we call attention back to our structural Assumption 1 for BAMDPs and note that, under the standard parameterization of epistemic

state via count parameters for Dirichlet priors/posteriors, we would only be able to assess residual epistemic uncertainty through differential entropy which, unlike the traditional (Shannon) entropy $\mathbb{H}(\cdot)$, is potentially negative and has no constant lower bound [Cover and Thomas, 2012].[2] Naturally, to compute the *information horizon of the BAMDP*, we need only take the supremum across the non-stationary policy class: $\mathcal{I} = \sup_{\pi \in \Pi^H} \mathcal{I}(\pi)$, where $\Pi = \{\mathcal{X} \to \mathcal{A}\}$.

Clearly, when it exists, we have that $1 \leq \mathcal{I} \leq H$; the case where $\mathcal{I} = 1$ corresponds to having a prior $p(\theta)$ that is itself a Dirac delta $\delta_\theta$ centered around the true environment, in which case, $\theta$ is known completely and the agent may simply compute and deploy the optimal policy for the MDP $\langle \mathcal{S}, \mathcal{A}, \mathcal{R}, \mathcal{T}_\theta, \beta, H \rangle$. At the other end of the spectrum, an information horizon $\mathcal{I} = H$ suggests that, in the worst case, an agent may need all $H$ steps of behavior in order to fully identify the environment. In the event that there exists any single non-stationary policy $\pi$ for which the infimum of $\mathcal{I}(\pi)$ does not exist (that is, $\mathcal{I}(\pi) = \infty$), then clearly $\mathcal{I} = \infty$; this represents the most difficult, worst-case scenario wherein an agent may not always capable of fully resolving its epistemic uncertainty within the specified problem horizon $H$. For certain scenarios, the supremum taken over the entire non-stationary policy class may be exceedingly strict and, certainly, creates a computational intractability should one wish to operationalize the information horizon algorithmically; in these situations, it may be more natural to consider smaller or regularized policy classes (for instance, the collection of expressible policies under a chosen neural network architecture) that yield more actionable notions of BAMDP complexity. We now go on to show how the information horizon can be used to design a more efficient BAMDP planning algorithm whose planning complexity bears a more favorable dependence on $H$.

### 4.3 Informed Value Iteration

The key insight from the previous section is that, when the information horizon exists and once an agent has acted for $\mathcal{I}$ timesteps, the Bayes-optimal policy necessarily falls back to the optimal policy associated with the true environment. Consequently, if the solutions to all $|\Theta|$ possible underlying MDPs are computed up front, an agent can simply backup their optimal values starting from the $\mathcal{I}$th timestep, rather than backing up values beginning at the original horizon $H$. This high-level idea is implemented as Algorithm 2 which assumes access to a sub-routine `mdp_value_iteration` that consumes a MDP and produces the associated optimal value function for the initial timestep, $V_1^\star$.

Since the underlying unknown MDP is one of $|\Theta|$ possible MDPs, Algorithm 2 proceeds by first computing the optimal value function associated with each of them in sequence using standard value iteration, incurring a time complexity of $\mathcal{O}(|\Theta||\mathcal{S}|^2|\mathcal{A}|(H - \mathcal{I}))$. Note that the horizon of each MDP is reduced to $H - \mathcal{I}$ acknowledging that, after identifying the true MDP in $\mathcal{I}$ steps, an agent has only $H - \mathcal{I}$ steps of interaction remaining with the environment. With these $|\Theta|$ solutions in hand, the remainder of the algorithm proceeds with standard value iteration for BAMDPs (as in Algorithm 1), only now bootstrapping value from the $\mathcal{I}$ timestep, rather than the original problem horizon $H$. Note that in Line 9, we could also compute the corresponding $\widehat{\theta}$ in question by taking the mean of the next epistemic state $p'$, however, we use this calculation to make explicit the fact that, by definition of the information horizon, the agent has no uncertainty in $\theta$ at this point. As a result, instead of planning complexity that scales the hyperstate space size by a potentially large problem horizon, we incur a complexity of $\mathcal{O}\left(|\Theta||\mathcal{S}||\mathcal{A}|\left(|\mathcal{X}|\mathcal{I} + |\mathcal{S}|(H - \mathcal{I})\right)\right)$. Naturally, as the gap between the information horizon $\mathcal{I}$ and problem horizon $H$ increases, the more favorably Algorithm 2 performs relative to the standard value iteration procedure of Algorithm 1.

In this section, we've demonstrated how the information horizon of a BAMDP has the potential to dramatically reduce the computational complexity of planning. Still, however, the corresponding guarantee bears an unfavorable dependence on the size of the hyperstate space $\mathcal{X}$ which, in the reality that voids Assumption 2, still renders both Algorithms 1 and 2 as computationally intractable. Since this is likely inescapable for the problem of computing the exact optimal BAMDP value function, the next section considers one path for reducing this burden at the cost of only being able to realize an approximately-optimal value function.

---

[2]Prior work (see, for example, Theorem 1 of Kolter and Ng [2009]) operating with the Dirichlet parameterization will make an alternative assumption for similar effect where epistemic state updates cease after a certain number of state-action pair visitations.

# 5 Epistemic State Abstraction

## 5.1 State Abstraction in MDPs

As numerous sample-efficiency guarantees in reinforcement learning [Kearns and Singh, 2002, Kakade et al., 2003b, Strehl et al., 2009] bear a dependence on the size of the MDP state space, $|\mathcal{S}|$, a large body of work has entertained state abstraction as a tool for improving the dependence on state space size without compromising performance [Whitt, 1978, Bertsekas et al., 1988, Singh et al., 1995, Gordon, 1995, Tsitsiklis and Van Roy, 1996, Dean and Givan, 1997, Ferns et al., 2004, Jong and Stone, 2005, Li et al., 2006, Van Roy, 2006, Ferns et al., 2012, Jiang et al., 2015a, Abel et al., 2016, 2018, 2019, Dong et al., 2019, Du et al., 2019, Misra et al., 2020, Abel, 2020]. Broadly speaking, a state abstraction $\phi : \mathcal{S} \to \mathcal{S}_\phi$ maps original or ground states of the MDP into abstract states in $\mathcal{S}_\phi$. Typically, one takes $\phi$ to be defined with respect to an abstract state space $\mathcal{S}_\phi$ with smaller complexity (in some sense) than $\mathcal{S}$; in the case of state aggregation where all spaces in question are finite, this desideratum often takes the very simple form of $|\mathcal{S}_\phi| < |\mathcal{S}|$. Various works have identified conditions under which specific classes of state abstractions $\phi$ yield no approximation error and perfectly preserve the optimal policy of the original MDP [Li et al., 2006], as well as conditions under which near-optimal behavior is preserved [Van Roy, 2006, Abel et al., 2016]. As its name suggests, our proposed notion of epistemic state abstraction aims to lift these kinds of guarantees for MDPs over to BAMDPs and contend with the intractably large hyperstate space.

Before examining BAMDPs, we provide a brief overview of how state abstraction impacts the traditional MDP, as a point of comparison with the BAMDP setting. Given a MDP $\langle \mathcal{S}, \mathcal{A}, \mathcal{R}, \mathcal{T}, \beta, H \rangle$, a state abstraction $\phi : \mathcal{S} \to \mathcal{S}_\phi$ induces a new abstract MDP $\mathcal{M}_\phi = \langle \mathcal{S}_\phi, \mathcal{A}, \mathcal{R}_\phi, \mathcal{T}_\phi, H \rangle$ where the abstract reward function $\mathcal{R}_\phi : \mathcal{S}_\phi \times \mathcal{A} \to [0,1]$ and transition function $\mathcal{T}_\phi : \mathcal{S}_\phi \times \mathcal{A} \to \Delta(\mathcal{S}_\phi)$ are both defined with respect to a fixed, arbitrary weighting function $\omega : \mathcal{S} \to [0,1]$ that, intuitively, measures the contribution of each individual MDP state $s \in \mathcal{S}$ to its allocated abstract state $\phi(s)$. More specifically, $\omega$ is required to induce a probability distribution on the constituent MDP states of each abstract state: $\forall s_\phi \in \mathcal{S}_\phi, \sum\limits_{s \in \phi^{-1}(s_\phi)} \omega(s) = 1$. This fact allows for well-defined rewards and transition probabilities as given by

$$\mathcal{R}_\phi(s_\phi, a) = \sum_{s \in \phi^{-1}(s_\phi)} \mathcal{R}(s, a)\omega(s), \qquad \mathcal{T}_\phi(s'_\phi \mid s_\phi, a) = \sum_{s \in \phi^{-1}(s_\phi)} \sum_{s' \in \phi^{-1}(s'_\phi)} \mathcal{T}(s' \mid s, a)\omega(s).$$

As studied by Van Roy [2006], the weighting function $\omega$ does bear implications on the efficiency of learning and planning. Naturally, one may go on to apply various planning or reinforcement-learning algorithms to $\mathcal{M}_\phi$ and induce behavior in the original MDP $\mathcal{M}$ by first applying $\phi$ to the current state $s \in \mathcal{S}$ and then leveraging the optimal abstract policy or abstract value function of $\mathcal{M}_\phi$. Conditions under which $\phi$ will induce a MDP $\mathcal{M}_\phi$ that preserves optimal or near-optimal behavior are studied by Li et al. [2006], Van Roy [2006], Abel et al. [2016].

## 5.2 Compressing the Epistemic State Space

In this section, we introduce epistemic state abstraction for BAMDPs with the goal of paralleling the benefits of state abstraction in MDPs. In particular, we leverage the fact that our epistemic state space $\Delta(\Theta) = \Delta^{|\Theta|-1}$ is the $(|\Theta| - 1)$-dimensional probability simplex. Recall that for any set $\mathcal{Z}$; any threshold parameter $\delta > 0$; and any metric $\rho : \mathcal{Z} \times \mathcal{Z} \to \mathbb{R}^+$ on $\mathcal{Z}$, a set $\{z_1, z_2, \ldots, z_K\}$ is a $\delta$-cover of $\mathcal{Z}$ if $\forall z \in \mathcal{Z}, \exists i \in [K]$ such that $\rho(z, z_i) \leq \delta$. In this work, we will consider $\delta$-covers with arbitrary parameter $\delta > 0$ defined on the simplex $\Delta^{|\Theta|-1}$ with respect to the total variation distance metric on probability distributions, denoted $|| \cdot ||_{\mathrm{TV}}$. Let $e_i \in \Delta(\Theta)$ be the $i$th standard basis vector such that $\mathbb{H}(e_i) = 0, \forall i \in [|\Theta|]$. We define an *epistemic state abstraction* with parameter $\delta > 0$ as the projection from $\Delta(\Theta)$ onto the smallest $\delta$-cover of $\Delta(\Theta)$ with respect to $|| \cdot ||_{\mathrm{TV}}$ that contains all standard basis vectors $\{e_1, e_2, \ldots, e_{|\Theta|}\}$; paralleling notation for the $\delta$-covering number, we use $\mathcal{N}(\Delta(\Theta), \delta, || \cdot ||_{\mathrm{TV}})$ to denote the size of this minimal cover and, for consistency with the state-abstraction literature in MDPs, use $\phi : \Delta(\Theta) \to \Delta_\phi(\Theta)$ to denote the epistemic state abstraction. Briefly, we note that while computing exact $\delta$-covers is a NP-hard problem, approximation algorithms do exist [Hochbaum, 1996, Zhang et al., 2012]; our work here is exclusively concerned with establishing theoretical guarantees that warrant further investigation of such approximation techniques to help solve BAMDPs in practice.

It is important to note that while there are numerous statistical results expressed in terms of covering numbers (for instance, Dudley's Theorem [Dudley, 1967]), our definition of covering number differs slightly in its inclusion of the standard basis vectors. The simple reason for this constraint is that it ensures we may still count on the existence of abstract epistemic states for which an agent has fully exhausted all epistemic uncertainty in the underlying environment. Consequently, we are guaranteed that the information horizon is still a well-defined quantity under this abstraction[3]. As $\delta$ increases, larger portions of the epistemic state space where the agent has residual, but still non-zero, epistemic uncertainty will be immediately mapped to the nearest standard basis vector under $\phi$. If such a lossy compression is done too aggressively, the agent's beliefs over the uncertain environment may prematurely and erroneously converge. On the other hand, if done judiciously with a prudent setting of $\delta$, one has the potential to dramatically reduce the complexity of planning across a much smaller, *finite* hyperstate space and recover an approximately-optimal BAMDP value function.

To make this intuition more precise, consider an initial BAMDP $\langle \mathcal{X}, \mathcal{A}, \overline{\mathcal{R}}, \overline{\mathcal{T}}, \overline{\beta}, H \rangle$ and, given an epistemic state abstraction $\phi : \Delta(\Theta) \to \Delta_\phi(\Theta)$ with fixed parameter $\delta > 0$, we recover an induced abstract BAMDP $\langle \mathcal{X}_\phi, \mathcal{A}, \overline{\mathcal{R}}_\phi, \overline{\mathcal{T}}_\phi, \overline{\beta}_\phi, H \rangle$ where, most importantly, $\mathcal{X}_\phi = \mathcal{S} \times \Delta_\phi(\Theta)$[4]. Just as in the MDP setting, the model of the abstract BAMDP depends on a fixed, arbitrary weighting function of the original epistemic states $\omega : \Delta(\Theta) \to [0, 1]$ that adheres to the constraint: $\forall p_\phi \in \Delta_\phi(\Theta)$, $\int_{\phi^{-1}(p_\phi)} \omega(p)dp = 1$, which means abstract rewards and transition probabilities for a current and next abstract hyperstates, $x_\phi = \langle s, p_\phi \rangle$ and $x'_\phi = \langle s', p'_\phi \rangle$, are given by

$$\overline{\mathcal{R}}_\phi(x_\phi, a) = \int_{\phi^{-1}(p_\phi)} \overline{\mathcal{R}}(x, a)\omega(p)dp = \int_{\phi^{-1}(p_\phi)} \mathcal{R}(s, a)\omega(p)dp = \mathcal{R}(s, a) \int_{\phi^{-1}(p_\phi)} \omega(p)dp = \mathcal{R}(s, a),$$

$$\overline{\mathcal{T}}_\phi(x'_\phi \mid x_\phi, a) = \int_{\phi^{-1}(p_\phi)} \omega(p) \sum_{p' \in \phi^{-1}(p'_\phi)} \overline{\mathcal{T}}(x' \mid x, a)dp, \text{ where } x = \langle s, p \rangle \text{ and } x' = \langle s', p' \rangle.$$

The initial abstract hyperstate distribution is defined as $\overline{\beta}_\phi = \beta \times \delta_{\phi(p(\theta))}$ where $\beta \in \Delta(\mathcal{S})$ denotes the initial state distribution of the underlying MDP while $\delta_{\phi(p(\theta))}$ is a Dirac delta centered around the agent's original prior, $p(\theta)$, projected by $\phi$ into the abstract epistemic state space. Observe that the abstract BAMDP transition function is stochastic with respect to the next abstract epistemic state $p'_\phi$, unlike the original BAMDP transition function whose next epistemic states are deterministic. This is, perhaps, not a surprising observation as it also occurs in standard state aggregation of deterministic MDPs as well. Nevertheless, it is important to note the corresponding abstract BAMDP value functions must now acknowledge this stochasticity for any abstract policy $\pi = (\pi_{\phi,1}, \pi_{\phi,2}, \ldots, \pi_{\phi,H})$, $\pi_{\phi,h} : \mathcal{X}_\phi \to \mathcal{A}, \forall h \in [H]$:

$$V^\pi_{\phi,h}(x_\phi) = Q^\pi_{\phi,h}(x_\phi, \pi_{\phi,h}(x_\phi)) \quad V^\star_{\phi,h}(x_\phi) = \max_{a \in \mathcal{A}} Q^\star_{\phi,h}(x_\phi, a), \qquad V^\pi_{\phi,H+1}(x_\phi) = 0 \quad V^\star_{\phi,H+1}(x_\phi) = 0 \quad \forall x_\phi \in \mathcal{X}_\phi,$$

$$Q^\pi_{\phi,h}(x_\phi, a) = \mathcal{R}(s, a) + \sum_{s', p'_\phi} \overline{\mathcal{T}}_\phi(x'_\phi \mid x_\phi, a)V^\pi_{\phi,h+1}(x'_\phi), \qquad Q^\star_{\phi,h}(x_\phi, a) = \mathcal{R}(s, a) + \sum_{s', p'_\phi} \overline{\mathcal{T}}_\phi(x'_\phi \mid x_\phi, a)V^\star_{\phi,h+1}(x'_\phi).$$

Beyond the fact that this abstract BAMDP enjoys a reduced hyperstate space, we further observe that the information horizon of this new BAMDP, $\mathcal{I}_\phi$, has the potential to be smaller than that of the original BAMDP. That is, if $\mathcal{I}$ steps are needed to fully resolve epistemic uncertainty in the original BAMDP then, by compressing the epistemic state space via $\phi$, we may find epistemic uncertainty exhausted in fewer than $\mathcal{I}$ timesteps within the abstract BAMDP. Furthermore, for a suitably large setting of the $\delta$ parameter, we also have cases where the original BAMDP has $\mathcal{I} = \infty$ while $\mathcal{I}_\phi < \infty$; in words, whereas it may not have been possible to resolve all epistemic uncertainty within $H$ timesteps, compression of the epistemic state space reduces this difficulty in the abstract problem as knowledge states near (in the total-variation sense) each vertex of the probability simplex $e_i$ are immediately aggregated. Due to space constraints, we defer further discussion of the abstract information horizon and its relationship with the original information horizon to Appendix C.

---

[3]Note that an alternative would be to introduce an additional constant $\gamma \in \mathbb{R}^+$ and define the information horizon based on $\mathbb{H}(p) \leq \gamma$; our construction avoids carrying this cumbersome additional parameter dependence in the results.

[4]One could also imagine abstracting over the original MDP state space $\mathcal{S}$ which, for clarity, we do not consider in this work.

As a toy illustration of last scenario, consider a $\phi$ with $\delta$ sufficiently large such that any step from the agent's prior distribution immediately maps to a next abstract hyperstate with no epistemic uncertainty. Clearly, regardless of $\mathcal{I}$, we have an abstract BAMDP where $\mathcal{I}_\phi = 2$. Of course, under such an aggressive abstraction, we should expect to garner an unfavorable degree of approximation error between the solutions of the abstract and original BAMDPs. The next section makes this error analysis and performance loss precise alongside an approximate planning algorithm that leverages the reduced complexity of abstract BAMDPs to recover a near-optimal solution to the original BAMDP of interest.

### 5.3 Informed Abstract Value Iteration

Observe that if, after inducing the abstract BAMDP according to a given epistemic state abstraction $\phi$, the resulting information horizon is finite $\mathcal{I}_\phi < \infty$, then we are in a position to run Algorithm 2 on the abstract BAMDP. Moreover, we no longer need the crutch of Assumption 2 as, by definition of $\phi$, we are guaranteed a finite abstract hyperstate space of size $|\mathcal{X}_\phi| = |\mathcal{S}| \cdot \mathcal{N}(\Delta(\Theta), \delta, || \cdot ||_{\mathrm{TV}})$. With the solution to the abstract BAMDP in hand, we can supply values to any input hyperstate of the original BAMDP $x = \langle s, p \rangle \in \mathcal{X}$ by simply applying $\phi$ to the agent's current epistemic state $p$ and querying the value of the resulting abstract hyperstate $\langle s, \phi(p) \rangle \in \mathcal{X}_\phi$. We present this approximate BAMDP planning procedure as Algorithm 3.

By construction, this algorithm inherits the planning complexity guarantee of Algorithm 2, specialized to the abstract BAMDP input, yielding $\mathcal{O}\left(|\Theta||\mathcal{S}|^2|\mathcal{A}|\left(\mathcal{N}(\Delta(\Theta), \delta, || \cdot ||_{\mathrm{TV}})^2\mathcal{I}_\phi + (H - \mathcal{I}_\phi)\right)\right)$. A key feature of this result is that we entirely forego a (direct) dependence on the hyperstate space of the original BAMDP and, instead, take on dependencies with the size of the abstract hyperstate space, $|\mathcal{X}_\phi|^2 = |\mathcal{S}|^2\mathcal{N}(\Delta(\Theta), \delta, || \cdot ||_{\mathrm{TV}})^2$, and the abstract information horizon $\mathcal{I}_\phi$. While both terms decrease as $\delta \to 1$, there is a delicate balance to be maintained between the ease with which one may solve the abstract BAMDP and the quality of the resulting solution when deployed in the original BAMDP of interest. We dedicate the remainder of this section to making this balance mathematically precise. Due to space constraints, all proofs are relegated to Appendix B. A natural first step in our analysis is to establish an approximation error bound:

**Proposition 1.** *Let $V_h^\star$ and $V_{\phi,h}^\star$ denote the optimal original and abstract BAMDP value functions, respectively, for any timestep $h \in [H]$. Let $\phi$ be an epistemic state abstraction as defined above. Then, $\max_{x \in \mathcal{X}} |V_h^\star(x) - V_{\phi,h}^\star(\phi(x))| \leq 2\delta(H - h)(H - h + 1)$.*

In order to establish a complimentary performance-loss bound, we require an intermediate result characterizing performance shortfall of a BAMDP value function induced by a greedy policy with respect to another near-optimal BAMDP value function. The analogue of this result for MDPs is proven by Singh and Yee [1994], and the proof for BAMDPs follows similarly.

**Proposition 2.** *Let $V = \{V_1, V_2, \ldots, V_H\}$ be an arbitrary BAMDP value function. We denote by $\pi_{h,V}$ the greedy policy with respect to $V$ defined $\forall x = \langle s, p \rangle \in \mathcal{X}$ as*

$$\pi_{h,V}(x) = \arg\max_{a \in \mathcal{A}} \left[\mathcal{R}(x, a) + \sum_{\theta, s'} \mathcal{T}_\theta(s' \mid s, a)p(\theta)V_{h+1}(x')\right],$$

*where $x' = \langle s', \mathcal{B}(p, s, a, s') \rangle \in \mathcal{X}$. Recall that $V_{h+1}^\star$ denotes the optimal BAMDP value function at timestep $h + 1$ and $\pi_h^\star$ denote the Bayes-optimal policy. If for all $h \in [H]$, for all $s \in \mathcal{S}$, and for any $p, q \in \Delta(\Theta)$ $|V_h^\star(\langle s, p \rangle) - V_h(\langle s, q \rangle)| \leq \varepsilon$, then $||V_h^\star - V_h^{\pi_{h,V}}||_\infty \leq 2\varepsilon(H - h + 1)$.*

Combining Propositions 1 and 2 immediately yields a corresponding performance-loss bound as desired, paralleling the analogous result for state aggregation in MDPs (see Theorem 4.1 of Van Roy [2006]):

**Proposition 3.** *Let $\pi_{\phi,h}^\star$ denote the greedy policy with respect to $V_{\phi,h+1}^\star$. Then, $||V_h^\star - V_h^{\pi_{\phi,h}^\star}||_\infty \leq 4\delta(H - h)(H - h + 1)^2$.*

## 6 Discussion & Conclusion

In this work, we began by characterizing the complexity of a BAMDP via an upper bound on the total number of interactions needed by an agent to exhaust information and fully resolve its

epistemic uncertainty over the true environment. Under an assumption on the exact form of the agent's uncertainty, we showed how this information horizon facilitates more efficient planning when smaller than the original problem horizon. We recognize that Assumption 1 deviates from the traditional parameterization of uncertainty in the MDP transition function via the Dirichlet distribution [Poupart et al., 2006, Kolter and Ng, 2009] (sometimes also known as the flat Dirichlet-Multinomial or FDM model [Asmuth, 2013]). The driving force behind this choice is to avoid dealing in differential entropy when engaging with the (residual) uncertainty contained in any epistemic state. Should one aspire to depart from Assumption 1 altogether in a rigorous way that manifests within the analysis, we suspect that it may be fruitful to consider a lossy compression of each epistemic state into a discrete, $|\Theta|$-valued random variable. Under such a formulation, the appropriate tool from information theory for the analysis would be rate-distortion theory [Cover and Thomas, 2012, Csiszár, 1974]. This would, in a theoretically-sound way, allow for an arbitrary BAMDP parameterization and, for the purposes of continuing the use (discrete) Shannon entropy in the definition of the information horizon, induce a lossy compression of each epistemic state whose approximation error relative to the true epistemic state could be accounted for via the associated rate-distortion function.

Recognizing the persistence of the intractable BAMDP hyperstate space, we then proceeded to outline epistemic state abstraction as a mechanism that not only induces a finite, tractable hyperstate space but also has the potential to incur a reduced information horizon within the abstract problem. Through our analysis of approximation error and performance loss, we observe an immediate consequence of Proposition 3: if one wishes to compute an $\varepsilon$-optimal BAMDP value function for an original BAMDP of interest, one need only find the $\frac{\varepsilon}{4(H-h)(H-h+1)^2}$-cover of the simplex, $\Delta(\Theta)$, and then apply the corresponding epistemic state abstraction through Algorithm 3, whose planning complexity bears no dependence on the hyperstate space of the original BAMDP and has reduced dependence on the problem horizon. One might observe that the right-hand side of the value-loss bound is maximized at timestep $h = 1$, making this first step the limiting factor when determining what value of $\delta$ to employ for computing the epistemic state abstraction. As this cover could become quite large and detract from the efficiency of utilizing an epistemic state abstraction in subsequent time periods, future work might benefit from considering abstractions formed by a sequence of exactly $H$ $\delta_h$-covers, where the indexing of $\delta_h$ in time $h \in [H]$ affords better preservation of value (via a straightforward extension of Proposition 3) across all timesteps simultaneously.

One caveat and limitation of our planning algorithms (both exact and approximate) is the provision of the information horizon as an input. An agent designer may seldom have the prescience of knowing the underlying BAMDP information structure or, even with suitable regularity assumptions on the policy class, be able to compute it. An observation is that many sampling-based algorithms for approximately solving BAMDPs, like BAMCP [Guez et al., 2012], implicitly hypothesize a small information horizon (typically, a value of 1) through their use of posterior sampling and choice of rollout policy. Meanwhile, recent work has demonstrated strong performance guarantees for a reinforcement-learning agent acting in an arbitrary environment [Dong et al., 2022] through the use of an incrementally increasing discount factor [Jiang et al., 2015b, Arumugam et al., 2018], gradually expanding the effective range over which the agent is expected to demonstrate competent behavior. Taking inspiration from this idea, future work might consider designing more-efficient planning algorithms that, while ignorant of the true information horizon, instead hypothesize a sequence of increasing information horizons, eventually building up to the complexity of the full BAMDP. Of course, prior to development of novel algorithms, the notion that existing BAMDP planners may already make implicit use of the information horizon is in and of itself a task for future work to tease apart and make mathematically rigorous.

Moreover, similar to how the simulation lemma [Kearns and Singh, 2002] provides a principled foundation for model-based reinforcement learning, our analysis might also be seen as offering theoretical underpinnings to the Bayes-optimal exploration strategies learned by meta reinforcement-learning agents [Ortega et al., 2019, Mikulik et al., 2020] whose practical instantiations already rely upon approximate representations of epistemic state [Zintgraf et al., 2019, 2021].

## Acknowledgements

The authors gratefully acknowledge the anonymous reviewers for their insightful comments, questions, and discussions.

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
