# A Algorithms

Here we present all algorithms discussed in the main paper.

---

**Algorithm 1** Value Iteration for BAMDPs

---

1: **Input:** BAMDP $\langle \mathcal{X}, \mathcal{A}, \overline{\mathcal{R}}, \overline{\mathcal{T}}, \overline{\beta}, H \rangle$
2: $V_{H+1}^\star(\langle s, p \rangle) = 0, \forall \langle s, p \rangle \in \mathcal{X}$
3: **for** $h = H, H-1, \ldots, 1$ **do**
4:    **for** $\langle s, p \rangle \in \mathcal{X}$ **do**
5:       **for** $a \in \mathcal{A}$ **do**
6:          $Q_h^\star(\langle s, p \rangle, a) = \mathcal{R}(s, a) + \sum_{s', \theta} \mathcal{T}_\theta(s' \mid s, a) p(\theta) V_{h+1}^\star(\langle s', \mathcal{B}(p, s, a, s') \rangle)$
7:       **end for**
8:       $V_h^\star(\langle s, p \rangle) = \max_{a \in \mathcal{A}} Q_h^\star(\langle s, p \rangle, a)$
9:    **end for**
10: **end for**

---

---

**Algorithm 2** Informed Value Iteration for BAMDPs

---

1: **Input:** BAMDP $\langle \mathcal{X}, \mathcal{A}, \overline{\mathcal{R}}, \overline{\mathcal{T}}, \overline{\beta}, H \rangle$, Information horizon $\mathcal{I} < \infty$
2: **for** $\theta \in \Theta$ **do**
3:    $V_\theta^\star = \texttt{mdp\_value\_iteration}(\langle \mathcal{S}, \mathcal{A}, \mathcal{R}, \mathcal{T}_\theta, \beta, H - \mathcal{I} \rangle)$
4: **end for**
5: **for** $h = \mathcal{I} - 1, \mathcal{I} - 2, \ldots, 1$ **do**
6:    **for** $\langle s, p \rangle \in \mathcal{X}$ **do**
7:       **for** $a \in \mathcal{A}$ **do**
8:          **if** $h + 1 == \mathcal{I}$ **then**
9:             **for** $s' \in \mathcal{S}$ **do**
10:               $p' = \mathcal{B}(p, s, a, s')$
11:               $\widehat{\theta} = \sum_{\theta \in \Theta} \theta \mathbb{1}(p'(\theta) = 1)$
12:               $V_{h+1}^\star(\langle s', p' \rangle) = V_{\widehat{\theta}}^\star$
13:             **end for**
14:          **end if**
15:          $Q_h^\star(\langle s, p \rangle, a) = \mathcal{R}(s, a) + \sum_{s', \theta} \mathcal{T}_\theta(s' \mid s, a) p(\theta) V_{h+1}^\star(\langle s', \mathcal{B}(p, s, a, s') \rangle)$
16:       **end for**
17:       $V_h^\star(\langle s, p \rangle) = \max_{a \in \mathcal{A}} Q_h^\star(\langle s, p \rangle, a)$
18:    **end for**
19: **end for**

---

**Algorithm 3** Informed Abstract Value Iteration for BAMDPs

---

1: **Input:** BAMDP $\langle \mathcal{X}, \mathcal{A}, \overline{\mathcal{R}}, \overline{\mathcal{T}}, \overline{\beta}, H \rangle$, Epistemic state abstraction $\phi$
2: Induce abstract BAMDP $\mathcal{M}_\phi = \langle \mathcal{X}_\phi, \mathcal{A}, \overline{\mathcal{R}}_\phi, \overline{\mathcal{T}}_\phi, \overline{\beta}_\phi, H \rangle$ with abstract information horizon
   $\mathcal{I}_\phi < \infty$
3: **for** $\theta \in \Theta$ **do**
4:    $V_\theta^\star = \texttt{mdp\_value\_iteration}(\langle \mathcal{S}, \mathcal{A}, \mathcal{R}, \mathcal{T}_\theta, \beta, H - \mathcal{I}_\phi \rangle)$
5: **end for**
6: **for** $h = \mathcal{I}_\phi - 1, \mathcal{I}_\phi - 2, \ldots, 1$ **do**
7:    **for** $\langle s, p_\phi \rangle \in \mathcal{X}_\phi$ **do**
8:       **for** $a \in \mathcal{A}$ **do**
9:          **if** $h + 1 == \mathcal{I}_\phi$ **then**
10:             **for** $\langle s', p'_\phi \rangle \in \mathcal{X}_\phi$ **do**
11:                $\widehat{\theta} = \sum_{\theta \in \Theta} \theta \mathbb{1}(p'_\phi(\theta) = 1)$
12:                $V_{\phi,h+1}^\star(\langle s', p'_\phi \rangle) = V_{\widehat{\theta}}^\star$
13:             **end for**
14:          **end if**
15:          $Q_{\phi,h}^\star(\langle s, p_\phi \rangle, a) = \mathcal{R}(s, a) + \sum_{s', p'_\phi} \overline{\mathcal{T}}_\phi(\langle s', p'_\phi \rangle \mid \langle s, p_\phi \rangle, a) V_{\phi,h+1}^\star(\langle s', p'_\phi \rangle)$
16:       **end for**
17:       $V_{\phi,h}^\star(\langle s, p_\phi \rangle) = \max_{a \in \mathcal{A}} Q_{\phi,h}^\star(\langle s, p_\phi \rangle, a)$
18:    **end for**
19:    For any $\langle s, p \rangle \in \mathcal{X}$, $V_h^\star(\langle s, p \rangle) = V_{\phi,h}^\star(\langle s, \phi(p) \rangle)$
20: **end for**

---

## B  Proofs

### B.1  Proof of Proposition 1

**Proposition 1.** *Let $V_h^\star$ and $V_{\phi,h}^\star$ denote the optimal original and abstract BAMDP value functions, respectively, for any timestep $h \in [H]$. Let $\phi$ be an epistemic state abstraction as defined above. Then,*

$$\max_{x \in \mathcal{X}} |V_h^\star(x) - V_{\phi,h}^\star(\phi(x))| \leq 2\delta(H - h)(H - h + 1).$$

*Proof.* With a slight abuse of notation, for any hyperstate $x \in \mathcal{X}$, let $\phi(x) = \langle s, p_\phi \rangle \in \mathcal{X}_\phi$ denote its corresponding abstract hyperstate where $p_\phi = \phi(p) \in \Delta_\phi(\Theta)$. For brevity, we define $p' \triangleq \mathcal{B}(p, s, a, s')$. We have

$$
\begin{aligned}
\max_{x \in \mathcal{X}} |V_h^\star(x) - V_{\phi,h}^\star(\phi(x))| &= \max_{\langle s, p \rangle \in \mathcal{X}} \left| \max_{a \in \mathcal{A}} Q_h^\star(\langle s, p \rangle, a) - \max_{a \in \mathcal{A}} Q_{\phi,h}^\star(\langle s, p_\phi \rangle, a) \right| \\
&\leq \max_{\langle s, p \rangle, a \in \mathcal{X} \times \mathcal{A}} |Q_h^\star(\langle s, p \rangle, a) - Q_{\phi,h}^\star(\langle s, p_\phi \rangle, a)| \\
&= \max_{\langle s, p \rangle, a} \left| \sum_{\theta, s'} \mathcal{T}_\theta(s' \mid s, a) p(\theta) V_{h+1}^\star(\langle s', p' \rangle) - \sum_{s', p'_\phi} \overline{\mathcal{T}}_\phi(\langle s', p'_\phi \rangle \mid \langle s, p_\phi \rangle, a) V_{\phi,h+1}^\star(\langle s', p'_\phi \rangle) \right|
\end{aligned}
$$

We now leverage the standard trick of adding "zero" by subtracting and adding the following between our two terms before applying the triangle inequality to separate them:

$$\sum_{s', p'_\phi} \overline{\mathcal{T}}_\phi(\langle s', p'_\phi \rangle \mid \langle s, p_\phi \rangle, a) V_{h+1}^\star(\langle s', p' \rangle).$$

Examining the first term in isolation, we first observe that, by definition of the weighting function, $\int_{\phi^{-1}(p_\phi)} \omega(\overline{p}) d\overline{p} = 1$ and so we have

$$\sum_{\theta, s'} \mathcal{T}_\theta(s' \mid s, a) p(\theta) V_{h+1}^\star(\langle s', p' \rangle) = \int_{\phi^{-1}(p_\phi)} \omega(\overline{p}) \sum_{\theta, s'} \mathcal{T}_\theta(s' \mid s, a) p(\theta) V_{h+1}^\star(\langle s', p' \rangle) d\overline{p}.$$

Expanding with the definition of the abstract BAMDP transition function, we have

$$\sum_{s',p'_\phi} \overline{\mathcal{T}}_\phi(\langle s',p'_\phi\rangle \mid \langle s,p_\phi\rangle, a)V^\star_{h+1}(\langle s',p'\rangle) = \int_{\phi^{-1}(p_\phi)} \omega(\overline{p})\sum_{\theta,s'}\mathcal{T}_\theta(s' \mid s,a)\overline{p}(\theta)V^\star_{h+1}(\langle s',p'\rangle)\sum_{p'_\phi}\mathbb{1}\left(\mathcal{B}(\overline{p},s,a,s') \in \phi^{-1}(p'_\phi)\right)$$

$$= \int_{\phi^{-1}(p_\phi)} \omega(\overline{p})\sum_{\theta,s'}\mathcal{T}_\theta(s' \mid s,a)\overline{p}(\theta)V^\star_{h+1}(\langle s',p'\rangle)\sum_{p'_\phi}\mathbb{1}\left(\phi(\mathcal{B}(\overline{p},s,a,s')) = p'_\phi\right)d\overline{p}$$

$$= \int_{\phi^{-1}(p_\phi)} \omega(\overline{p})\sum_{\theta,s'}\mathcal{T}_\theta(s' \mid s,a)\overline{p}(\theta)V^\star_{h+1}(\langle s',p'\rangle)d\overline{p},$$

since $\phi(\mathcal{B}(\overline{p},s,a,s'))$ belongs to exactly one abstract epistemic state. Using the fact that $V^\star_{h+1}(\langle s',p'\rangle) \leq H - h$ and simplifying, we have

$$\left|\sum_{\theta,s'}\mathcal{T}_\theta(s' \mid s,a)p(\theta)V^\star_{h+1}(\langle s',p'\rangle) - \int_{\phi^{-1}(p_\phi)} \omega(\overline{p})\sum_{\theta,s'}\mathcal{T}_\theta(s' \mid s,a)\overline{p}(\theta)V^\star_{h+1}(\langle s',p'\rangle)d\overline{p}\right|$$

$$\leq (H-h)\int_{\phi^{-1}(p_\phi)} \omega(\overline{p})\sum_\theta |p(\theta) - \overline{p}(\theta)|d\overline{p}$$

$$= (H-h)\int_{\phi^{-1}(p_\phi)} \omega(\overline{p})2\cdot\frac{1}{2}\sum_\theta |p(\theta) - \overline{p}(\theta)|d\overline{p}$$

$$= (H-h)\int_{\phi^{-1}(p_\phi)} \omega(\overline{p})2\cdot ||p(\theta) - \overline{p}(\theta)||_{\mathrm{TV}}d\overline{p}$$

$$\leq 4\delta(H-h)\int_{\phi^{-1}(p_\phi)} \omega(\overline{p})d\overline{p}$$

$$= 4\delta(H-h),$$

where the last upper bound follows from the definition of a $\delta$-cover since $\phi(p) = \phi(\overline{p}) = p_\phi$, $\forall \overline{p} \in \phi^{-1}(p_\phi)$.

Moving on to the second term and applying Jensen's inequality, we have

$$\left|\sum_{s',p'_\phi} \overline{\mathcal{T}}_\phi(\langle s',p'_\phi\rangle \mid \langle s,p_\phi\rangle, a)V^\star_{h+1}(\langle s',p'\rangle) - \sum_{s',p'_\phi} \overline{\mathcal{T}}_\phi(\langle s',p'_\phi\rangle \mid \langle s,p_\phi\rangle, a)V^\star_{\phi,h+1}(\langle s',p'_\phi\rangle)\right|$$

$$\leq \sum_{s',p'_\phi} \overline{\mathcal{T}}_\phi(\langle s',p'_\phi\rangle \mid \langle s,p_\phi\rangle, a)\left|V^\star_{h+1}(\langle s',p'\rangle) - V^\star_{\phi,h+1}(\langle s',p'_\phi\rangle)\right|$$

$$\leq \max_{x\in\mathcal{X}}|V^\star_{h+1}(x) - V^\star_{\phi,h+1}(\phi(x))|.$$

Thus, putting everything together, we have established that

$$\max_{x\in\mathcal{X}}|V^\star_h(x) - V^\star_{\phi,h}(\phi(x))| \leq 4\delta(H-h) + \max_{x\in\mathcal{X}}|V^\star_{h+1}(x) - V^\star_{\phi,h+1}(\phi(x))|$$

Iterating the same sequence of steps for the latter term on the right-hand side $H - h$ more times, we arrive at a final bound

$$\max_{x\in\mathcal{X}}|V^\star_h(x) - V^\star_{\phi,h}(\phi(x))| \leq \sum_{\overline{h}=h}^{H} 4\delta(H-\overline{h}) = 4\delta\sum_{\overline{h}=1}^{H-h}\overline{h} = 4\delta\frac{(H-h)(H-h+1)}{2} = 2\delta(H-h)(H-h+1).$$

$$\square$$

## B.2 Proof of Proposition 2

**Proposition 2.** *Let $V = \{V_1, V_2, \ldots, V_H\}$ be an arbitrary BAMDP value function. We denote by $\pi_{h,V}$ the greedy policy with respect to $V$ defined as*

$$\pi_{h,V}(x) = \arg\max_{a \in \mathcal{A}} \left[ \mathcal{R}(x,a) + \sum_{\theta, s'} \mathcal{T}_\theta(s' \mid s, a) p(\theta) V_{h+1}(x') \right] \qquad \forall x = \langle s, p \rangle \in \mathcal{X},$$

*where $x' = \langle s', \mathcal{B}(p, s, a, s') \rangle \in \mathcal{X}$. Recall that $V_{h+1}^\star$ denotes the optimal BAMDP value function at timestep $h+1$ and $\pi_h^\star$ denote the Bayes-optimal policy. If for all $h \in [H]$, for all $s \in \mathcal{S}$, and for any $p, q \in \Delta(\Theta)$*

$$|V_h^\star(\langle s, p \rangle) - V_h(\langle s, q \rangle)| \leq \varepsilon, \text{ then } ||V_h^\star - V_h^{\pi_{h,V}}||_\infty \leq 2\varepsilon(H - h + 1).$$

*Proof.* Fix an arbitrary timestep $h \in [H]$. For any $x \in \mathcal{X}$, define $a, \overline{a} \in \mathcal{A}$ such that $a = \pi_h^\star(x)$ and $\overline{a} = \pi_{h,V}(x)$. Similarly, let $p' = \mathcal{B}(p, s, a, s')$ and $\overline{p}' = \mathcal{B}(p, s, \overline{a}, s')$. Since, by definition, $\pi_{h,V}$ is greedy with respect to $V_{h+1}$, we have that

$$\mathcal{R}(x, a) + \sum_{\theta, s'} \mathcal{T}_\theta(s' \mid s, a) p(\theta) V_{h+1}(\langle s', p' \rangle) \leq \mathcal{R}(x, \overline{a}) + \sum_{\theta, s'} \mathcal{T}_\theta(s' \mid s, \overline{a}) p(\theta) V_{h+1}(\langle s', \overline{p}' \rangle).$$

By assumption, we have that

$$V_{h+1}^\star(\langle s', p' \rangle) - \varepsilon \leq V_{h+1}(\langle s', p' \rangle) \qquad V_{h+1}(\langle s', \overline{p}' \rangle) \leq V_{h+1}^\star(\langle s', p' \rangle) + \varepsilon.$$

Applying both bounds to the above yields

$$\mathcal{R}(x, a) + \sum_{\theta, s'} \mathcal{T}_\theta(s' \mid s, a) p(\theta)(V_{h+1}^\star(\langle s', p' \rangle) - \varepsilon) \leq \mathcal{R}(x, \overline{a}) + \sum_{\theta, s'} \mathcal{T}_\theta(s' \mid s, \overline{a}) p(\theta)(V_{h+1}^\star(\langle s', p' \rangle) + \varepsilon).$$

Consequently, we have that

$$\mathcal{R}(x, a) - \mathcal{R}(x, \overline{a}) \leq 2\varepsilon + \sum_\theta p(\theta) \sum_{s'} V_{h+1}^\star(\langle s', p' \rangle) \left[ \mathcal{T}_\theta(s' \mid s, \overline{a}) - \mathcal{T}_\theta(s' \mid s, a) \right].$$

From this, it follows that

$$\begin{aligned}
||V_h^\star - V_h^{\pi_{h,V}}||_\infty &= \max_{x \in \mathcal{X}} |V_h^\star(x) - V_h^{\pi_{h,V}}(x)| \\
&= \max_{x \in \mathcal{X}} |Q_h^\star(x, a) - Q_h^{\pi_{h,V}}(x, \overline{a})| \\
&= \max_{x \in \mathcal{X}} \left| \mathcal{R}(x, a) - \mathcal{R}(x, \overline{a}) + \sum_\theta p(\theta) \sum_{s'} \left[ \mathcal{T}_\theta(s' \mid s, a) V_{h+1}^\star(\langle s', p' \rangle) - \mathcal{T}_\theta(s' \mid s, \overline{a}) V_{h+1}^{\pi_{h+1,V}}(\langle s', \overline{p}' \rangle) \right] \right| \\
&\leq 2\varepsilon + \max_{\langle s, p \rangle} \left| \sum_\theta p(\theta) \sum_{s'} \mathcal{T}_\theta(s' \mid s, \overline{a}) \left[ V_{h+1}^\star(\langle s', p' \rangle) - V_{h+1}^{\pi_{h+1,V}}(\langle s', \overline{p}' \rangle) \right] \right| \\
&\leq 2\varepsilon + ||V_{h+1}^\star - V_{h+1}^{\pi_{h+1,V}}||_\infty \\
&\vdots \\
&\leq 2\varepsilon(H - h + 1).
\end{aligned}$$

where the last inequality follows by iterating the same procedure for the second term in the penultimate inequality across the remaining $H - h$ timesteps. $\square$

## B.3 Proof of Proposition 3

**Proposition 3.** *Let $\pi_{\phi,h}^\star$ denote the greedy policy with respect to $V_{\phi,h+1}^\star$. Then,*

$$||V_h^\star - V_h^{\pi_{\phi,h}^\star}||_\infty \leq 4\delta(H - h)(H - h + 1)^2.$$

*Proof.* Since, for any $x \in \mathcal{X}$, $\phi(x)$ differs only in the epistemic state, the proof follows by realizing that the $\varepsilon$ term of Proposition 2 is established by Proposition 1. Namely,

$$||V_h^\star - V_h^{\pi_{\phi,h}^\star}||_\infty \leq 2(H - h + 1) \max_{x \in \mathcal{X}} |V_h^\star(x) - V_{\phi,h}^\star(\phi(x))| \leq 4\delta(H - h)(H - h + 1)^2$$

$\square$

# C   On the Reduction of the Abstract Information Horizon

In this section, we offer two simple yet illustrative examples of BAMDPs where the use of epistemic state abstraction can either decrease or increase the information horizon of the resulting abstract BAMDP. Taken together with the main results of the paper, these examples underscore how epistemic state abstraction, similar to traditional state abstraction in MDPs, is not a panacea to sample-efficient BAMDP planning. Further investigation is needed to clarify the conditions under which an epistemic state abstraction may actually deliver upon the theoretical benefits outlined in this work.

## C.1   Decreased Information Horizon under Epistemic State Abstraction

Consider a MDP whose state space is defined on the non-negative integers $\mathcal{S} = \mathbb{Z}^+ = \{0, 1, 2, \ldots\}$ with two actions $\mathcal{A} = \{+, \circ\}$. For the purposes of this example, we ignore the reward function and focus on the transition function where, for any timestep $h \in [H]$, there is a fixed parameter $q > \frac{1}{2}$ such that $s_{h+1} = s_h + \Delta$ with $\Delta \sim \text{Bernoulli}(q)$ if $a_h = +$ and $\Delta \sim \text{Bernoulli}(1 - q)$ if $a_h = \circ$. In words, action $+$ has a higher probability of incrementing the agent's current state by one whereas the $\circ$ action is more likely to leave the agent's state unchanged.

For the corresponding BAMDP, we take $\Theta = \{\theta_1, \theta_2\}$ where $\theta_1$ corresponds to the true MDP transition function as described above. Meanwhile, $\theta_2$ simply corresponds to the transition function of $\theta_1$ with the actions flipped such that $\Delta \sim \text{Bernoulli}(q)$ if $a_h = \circ$ and $\Delta \sim \text{Bernoulli}(1 - q)$ if $a_h = +$. Clearly, when $q = 1$, epistemic uncertainty in this BAMDP is resolved immediately by the first transition whereas, with $q \downarrow \frac{1}{2}$, the two hypotheses become increasingly harder to distinguish, requiring more observed transitions from the environment and potentially exceeding the finite problem horizon $H$.

Now consider the epistemic state abstraction of $\Delta(\Theta)$ with parameter $\delta > 0$. Increasing $\delta \uparrow 1$ is commensurate with setting a $\gamma \in \mathbb{R}_{\geq 0}$ such that once the entropy of the current epistemic state falls below this threshold $\mathbb{H}(p_h) \leq \gamma$, we immediately have that $\phi(p_h) \in \{e_1, e_2\}$, where $e_i$ denotes the $i$th standard basis vector in $\Delta(\Theta)$. By construction, any action taken in this MDP necessarily reveals information to help reduce epistemic uncertainty. Consequently, with enough observed transitions from the environment, we can use $\phi$ to collapse the agent's beliefs around $\theta_1$ with far fewer samples than what would be needed to fully exhaust epistemic uncertainty. Moreover, depending on the exact problem horizon $H$, this could be used to recover a finite abstract information horizon from what was an infinite information horizon in the original BAMDP.

More concretely, suppose $H = 3$ and $q = \frac{4}{5}$. While any policy will quickly stumble upon $\theta_1$ as the most likely outcome, such a short horizon $H$ likely does not allow the entropy of all epistemic states to fall to zero, potentially yielding an infinite information horizon $\mathcal{I} = \infty$. However, there certainly exists a value of $\delta$ such that the first two steps of behavior under any policy is sufficient for reaching a vertex state in $\Delta_\phi(\Theta)$ and identifying the underlying MDP, ultimately yielding $\mathcal{I}_\phi = 2$.

## C.2   Increased Information Horizon under Epistemic State Abstraction

To show how epistemic state abstraction can work unfavorably and increase the information horizon, we consider a BAMDP where all policies rapidly resolve epistemic uncertainty, but the incorporation of an epistemic state abstraction may compromise the agent's ability to reach a vertex of the simplex $\Delta(\Theta)$.

For any $N \in \mathbb{N}$, consider a single initial state connected to two $N$-state chains (an upper chain and a lower chain) with two possible actions $\mathcal{A} = \{a_1, a_2\}$. For each possible $\theta \in \Theta$, the corresponding transition function $\mathcal{T}_\theta$ will either have $a_1$ deterministically transition to the next state in the upper chain and have $a_2$ send the agent to the next state in the lower chain, or vice versa. In other words, each transition function $\mathcal{T}_\theta$ can be concisely encoded as a length $N$ binary string $\in \{0, 1\}^N$ where, for each $n \in [N]$, the $n$th bit equal to 1 implies action $a_1$ transitions to the upper chain (immediately implying $a_2$ transitions to the lower chain) while a value of 0 signifies the opposite transition structure. Further suppose that, for each $n \in [N]$, the $n$th bit of all but one of the transition functions is identical; said differently, this structural assumptions says that, in the $n$th stage of the chain, there is a single informative transition that would uniquely identify the underlying MDP. A consequence of this is that, in the worst case, a BAMDP policy will only see one of the uninformative transitions in each

stage and, therefore, can only eliminate exactly one hypothesis from $\Theta$ with each timestep. Thus, for any problem horizon $H \geq N$, we are guaranteed an information horizon of $N$.

To see how epistemic state abstraction might inhibit planning, consider the policy that misses the informative transition in each stage. For an epistemic state abstraction with $\delta$ sufficiently large, the agent will remain stuck (via a self-looping transition) in the initial abstract epistemic state as no single uninformative transition yields sufficient information gain to move the agent through the abstract epistemic state space. This phenomenon of state abstraction ameliorating generalization while drastically worsening the challenge of exploration has already been observed in the standard MDP setting [Abel et al., 2017, 2020]; here, we see that epistemic state abstraction is also vulnerable to the same weakness. As a result, the agent will never converge to one of the vertices of the simplex and never see its epistemic uncertainty in the underlying environment completely diminished, resulting in $\mathcal{I}_\phi = \infty$.