# OpenReview forum: "Planning to the Information Horizon of BAMDPs via Epistemic State Abstraction"
_NeurIPS.cc/2022/Conference — NeurIPS 2022 Accept_

### Official Review · Reviewer_Gwj8 · 2022-06-28

**Rating:** 5
**Confidence:** 4
**Soundness:** 4 excellent
**Presentation:** 3 good
**Contribution:** 2 fair

**Summary:**

This paper introduces a novel concept for analyzing the complexity of Bayes-Adaptive MDPs (BAMDPs), which is called the Information Horizon. The authors argue that BAMDPs with low Information Horizon (relative to the episode horizon) -- that is, those BAMDPs in which an agent can potentially deduce the underlying MDP without uncertainty after relatively few environment steps -- can be solved more efficiently since value iteration in the hyperstate space is carried out over a shorter horizon. The authors then demonstrate how the complexity of (approximately) solving BAMDPs can be further reduced via state abstraction, and error bounds are provided for the value functions learned in such abstract BAMDPs.

**Questions:**

# Questions and Comments

**Section 2.1 -- Definition of the value function**: I believe there is
a typo in the state/action indexing in the sum -- it should be
$\mathcal{R}(s_{h'}, a_{h'})$ rather than $\mathcal{R}(s_h,a_h)$.

**Assumption 1**: Is the assumption that the parameter space $\Theta$ is
finite not much more limiting than existing work? In the setting with a
Dirichlet prior that is mentioned, the agent would be modeling a belief
over uncountably many MDPs. Moreover, on line 71, it says \"we do not
concern ourselves with the computation of the posterior\" -- but
doesn\'t this neglect much of the issue of intractability in BAMDPs?
Finally, how fair is it really to assume a constant-time posterior
update? Even with finitely-many MDPs, I would expect that the number of
known MDPs ($|\Theta|$) must somehow grow with the state/action spaces
in a reasonable model.

**Assumption 2**: I believe this is notationally incorrect. While you
assume \"an agressively-fine quantization\" of $\Delta(\Theta)$,
$\Delta(\Theta)$ itself is not finite, so $|\mathcal{X}|\not < \infty$.
The quantization of $\Delta(\Theta)$ should be made explicit, i.e.,
\"let $\widehat{\Delta}\subset\Delta(\Theta)$ be finite... let
$\mathcal{X}
= \mathcal{S}\times\widehat{\Delta}$\".

**Value iteration in hyperstate space**: When discussing the complexity
of value iteration in hyperstate space, the term $|\mathcal{X}|$ appears
frequently. To my understanding, this term hides a lot of complexity. In
particular, we\'re assuming an aggressively-fine quantization of
$\Delta(\Theta)$ as part of the space $\mathcal{X}$, which leads me to
believe that $|\mathcal{X}|\in
\mathcal{O}(|\mathcal{S}|\exp(|\Theta| - 1))$. If
$|\Theta|\in\omega(\log|\mathcal{S}|)$, then already any algorithm with
complexity linear in $|\mathcal{X}|$ is intractable. Conversely, if $|\Theta|\in \mathcal{O}(\log|\mathcal{S}|)$, the algorithm may actually be tractable. Thus, I think the
scaling of $|\Theta|$ w.r.t. $|\mathcal{S}|$ should be discussed. Note
that if $|\Theta|\in\mathcal{O}(1)$, then as
$\mathcal{S}\uparrow\infty$, one has less and less uncertainty over the
MDP. While this does appear to be addressed in line 222, it may be
interesting to discuss settings where these algorithms *are* tractable,
since already so many \"finiteness\" assumptions have been made.

**Proposition 1**: Naively, one should expect that
$\max_{x\in\mathcal{X}}|V_h^\star(x) -
V_{\phi,h}^\star(\phi(x))|\leq H - h + 1$ since the rewards are bounded
in $[0,1]$. For this proposition to be at all interesting, we must have
$\delta(H - h) < 1/2$. Thus, as the horizon increases, the parameter
$\delta$ must decrease proportionally for this to be a meaningful
result.

**Proposition 3**: Similar to my comment about Proposition 1: for this
result to be meaningful, we must have $\delta<(4(H - h)(H - h +
1))^{-1}$. Altogether, for this propositions to match the naive
performance-loss bound, we need $\delta\in O(1/H^2)$. This is addressed
on line 349, but can this be reflected in the complexity bound on line
312, via the
$\mathcal{N}(\Delta(\Theta), \delta, \|\cdot\|_{\text{TV}})$ term? I
suspect that the size of the cover would be very large in this case,
which would add greater dependence on $H$ to the complexity, which the
goal was to avoid.

# Minor issues

-   It would be helpful to have hyperlinks. In particular, the long list
    of citations starting on line 21 is very daunting when the only way
    to check the citations is by manually scrolling through the
    references section.
-   For the long list of citations starting on line 21, it would also be
    nice if they were listed in numerical order, which make it easier to
    scan the list without navigating back in forth between line 21 and
    the references section so many times.


**Limitations:**

As previously mentioned, I believe the theoretical results are quite limited due to the assumptions on the belief space over MDPs. The authors discuss these limitations briefly, although it would have been nice to see more in-depth discussion.

**Strengths And Weaknesses:**

# Strengths
The goal of this paper -- to identify a class of BAMDPs that can be solved efficiently -- is a very interesting and ambitious one. For the most part, the authors do a good job of clearly stating assumptions and precisely defining the setting that they're working in. The notion of the information horizon is quite novel to the best of my knowledge, and it seems to be a reasonable and important concept. The proofs of theoretical results are clear and appear to be correct.

# Weaknesses
I'm mainly concerned with some of the assumptions that are made throughout the paper. In particular, the assumption that the agent maintains a belief over a finite, constant number of MDPs seems excessively limiting. It would have at least been interesting to have a discussion about cases where the number of MDPs in the support of the agent's belief could increase with the size of the state space, even if it must remain finite. Consequently, I'm afraid the results in this paper may have limited relevance.

## Post-Rebuttal Update

The authors cleared up most of my concerns. I am still skeptical about the relevance of the problem setting, particularly that of a constant-size prior MDP support regardless of the size of the MDP, but I think the information horizon approach is interesting. I have adjusted my score to reflect this.

---

> ### Author Response · Authors · 2022-08-02
> **Response to Reviewer Gwj8 (1/2)**
>
> We appreciate the reviewer’s time and effort in providing feedback on our paper. We were glad to see that the reviewer recognizes the impact and value of our theoretical contributions.
>
> > Section 2.1 -- Definition of the value function: I believe there is a typo in the state/action indexing in the sum
>
> The reviewer is correct; thank you for pointing out the typo.
>
> > Assumption 2: I believe this is notationally incorrect. While you assume "an agressively-fine quantization"
>
> Again, the reviewer is correct and we agree that this is a more mathematically precise framing of the assumption. We will update it in the next iteration of the paper accordingly.
>
> > It would be helpful to have hyperlinks. In particular, the long list of citations starting on line 21 is very daunting when the only way to check the citations is by manually scrolling through the references section.
>
> We wholeheartedly agree that having backrefs in the citation list itself would be of great assistance to readers. We’ll happily include them in the next iteration of the paper assuming they are compatible with the NeurIPS style.
>
> > For the long list of citations starting on line 21, it would also be nice if they were listed in numerical order, which make it easier to scan the list without navigating back in forth between line 21 and the references section so many times.
>
> We appreciate the reviewer’s perspective on this, however, the references are currently ordered in a nearly chronological order, which we feel supersedes the aesthetic of the numerical order. This would especially be the case if an additional tenth page of a camera-ready submission would allow us to use the standard (author; year) citation style instead of the numbers style in natbib.
>
> > Is the assumption that the parameter space is finite not much more limiting than existing work? In the setting with a Dirichlet prior that is mentioned, the agent would be modeling a belief over uncountably many MDPs.
>
> Exactly, we felt it was important to formalize Assumption 1 precisely for the reason cited by the reviewer; namely, that the standard presentation of tabular BAMDPs would leverage a Dirichlet prior/posterior over each state-action pair, effectively enabling a belief over all possible transition functions (equivalently, all possible $|\mathcal{S}||\mathcal{A}| \times |\mathcal{S}|$ stochastic matrices). This is in contrast with Assumption 1 which precludes this exactly.
>
> > Moreover, on line 71, it says "we do not concern ourselves with the computation of the posterior" -- but doesn't this neglect much of the issue of intractability in BAMDPs? Finally, how fair is it really to assume a constant-time posterior update?
>
> Following the standard Dirichlet parameterization mentioned in the previous response for tabular MDPs, we feel that this is reasonable since posterior updating amounts to incrementing visitation counts by one upon observing any single transition. Similarly, posterior updates under the provision of Assumption 1 amount to re-weighting the prior probabilities for each of the $|\Theta|$ potential models according to the likelihood of generating each observed transition. For a fixed choice of $\Theta$, this update would amount to being constant time.

---

> > ### Author Response · Authors · 2022-08-02
> > **Response to Reviewer Gwj8 (2/2)**
> >
> > > Even with finitely-many MDPs, I would expect that the number of known MDPs () must somehow grow with the state/action spaces in a reasonable model.
> >
> > > Value iteration in hyperstate space: When discussing the complexity of value iteration in hyperstate space, the term appears frequently. To my understanding, this term hides a lot of complexity. In particular, we're assuming an aggressively-fine quantization of as part of the space , which leads me to believe that . If , then already any algorithm with complexity linear in is intractable. Conversely, if , the algorithm may actually be tractable. Thus, I think the scaling of w.r.t. should be discussed. Note that if , then as , one has less and less uncertainty over the MDP.
> >
> > We were initially confused by these comments from the reviewer which seem to suggest a formal link between the number of potential MDPs over which an agent represents its epistemic uncertainty and the size of the state or state-action space. After some consideration, we hypothesize that the reviewer is specifically considering a Dirichlet parameterization where $\Theta$ would incur a dependence on the state-action space as each transition distribution would carry the $|\mathcal{S}|$ parameters of a Dirichlet distribution. We recall, however, that Assumption 1 is specifically there to move us away from this parameterization in favor of categorical distributions whose support is over some known collection of potential environments. The only real functions of Assumption 1 are to bring finiteness to the hyperstate space (by virtue of allowing each epistemic state to be a point in the probability simplex) and to avoid the use of differential entropy (L185).  If we are incorrect about the reviewer’s specific consideration of a Dirichlet parameterization then, from our perspective, there doesn’t seem to be a reason why this structural assumption meant to impact the epistemic state space should carry a dependence on the cardinality of the original state space (or action space) of the underlying MDP.
> >
> >
> > > Proposition 1: Naively, one should expect that
> >
> > > Proposition 3: Similar to my comment about Proposition 1: for this result to be meaningful, we must have
> >
> > The reviewer is correct about the interplay between $\delta$ and the original problem horizon $H$. We believe that our presentation of both the planning complexity result and associated performance loss bound are standard with the reinforcement learning theory literature, in the sense that the true theoretical limits are $\delta \in (0,1)$ and, as with many theoretical guarantees, there are settings of the parameter whose associated performance guarantees are vacuous; value-loss bounds for state abstractions are another example of such guarantees.
> >
> > Notice that the term in question has $H - h$, rather than $H$ by itself. Implicit in the reviewer’s comment is consideration specifically for the first timestep $h=1$, where this quantity is of course maximized. So, perhaps, a takeaway the reviewer is hinting at here (which we would gladly find space for in the next iteration of the paper) is that, in some BAMDPs, a more efficient way of preserving value via an epistemic state abstraction across all timesteps is to consider a sequence of covers (indexed by $h \in [H]$); such an idea is supported by the penultimate line of the proof of Proposition 1 where a specific sequence of steps is unrolled inductively across successive value function differences. While being a straightforward extension of our analysis, the reviewer’s comment holds that, with a sufficiently large horizon $H$, such a step could be critical to ensuring the reduced dependence on problem horizon that the paper strives to maintain. We hope that this response accurately depicts the reviewer’s concern and thank them for raising it.

---

> > > ### Comment · Reviewer_Gwj8 · 2022-08-02
> > > **Clarifying**
> > >
> > > Thanks to the authors for such a detailed response, which cleared up some of my concerns. With regard to my question about value iteration in hyperstate space, I have a feeling there was a misunderstanding, and likely my original question was unclear. I will try to clarify that here.
> > >
> > > My point essentially is the following: for fixed state space $S$ and action space $A$, let $M(S, A)$ denote the set of all MDPs over these state and action spaces. Clearly, as $|S|$ increases, $|M(S, A)|$ increases (exponentially?), even if the transition kernels are deterministic. I understand that you're forcing the posteriors to have a finite support, but I suppose I'm trying to highlight the difference between "the posterior has finite support over a number of MDPs independent of $|S|$", and "for each finite $S$, the posterior is supported over $C(S)<\infty$ MDPs". To my understanding, your Assumption 1 assumes the former -- that is, you model a posterior over $N$ MDPs regardless of the size of $S$.
> > >
> > > Now, if indeed this is your assumption, my question is "is this a meaningful assumption"? Perhaps it is, but at least intuitively to me, I would find this assumption to be overly restrictive. For example, if you were to try to implement such a BAMDP algorithm when $|S| = 1000$, how would you even specify the support of the prior/posterior? You would need extremely specific information about what the possible MDPs are. Note that if the true MDP is not in the support of your prior, it cannot be in the support of the posterior, so the inferred MDP can be more and more dissimilar to the true MDP as the MDP space grows. I understand the principal contributions of this paper are theoretical, but I'm just trying to understand if this setting is actually one that the RL community is interested in. Given the other reviews, it appears that my judgement may have been wrong and that the community is in fact interested in this setting -- but can you point me to any work that investigates a setting with a similar assumption? Or are you arguing that this is a novel setting that RL theory should focus on?
> > >
> > > On the other hand, if Assumption 1 is assuming that the support of the posterior can be a function of $S$, I would argue that the hyperstate space grows at least exponentially with $|S|$, so writing bounds in terms of the size of the hyperstate space may be hiding an exponential factor.

---

> > > > ### Author Response · Authors · 2022-08-04
> > > > **Reply to Reviewer Gwj8 (1/2)**
> > > >
> > > > We thank the reviewer for their clarifying comment.
> > > >
> > > > > To my understanding, your Assumption 1 assumes the former -- that is, you model a posterior over $N$ MDPs regardless of the size of $S$.
> > > >
> > > > Yes, the reviewer is correct with this interpretation of Assumption 1 and that we do not maintain a link between the size of the underlying MDP and the number of MDPs which fall under the support of the prior/posterior.
> > > >
> > > > > Now, if indeed this is your assumption, my question is "is this a meaningful assumption"? Perhaps it is, but at least intuitively to me, I would find this assumption to be overly restrictive. For example, if you were to try to implement such a BAMDP algorithm when $|S| = 1000$, how would you even specify the support of the prior/posterior? You would need extremely specific information about what the possible MDPs are.
> > > >
> > > > Again, the reviewer is correct that this assumption would require one to have exact knowledge of each potential MDP so as to be able to compute posterior updates (that is, each candidate MDP model over the state space with $|\mathcal{S}| = 1000$). Notice, however, that this must be true of any parameterization in order to compute likelihoods of observed transitions. Since $\Theta$ is finite, the support can simply be taken as the corresponding index set $[|\Theta|]$ and, when the time eventually arrives to perform posterior updates, some fixed but arbitrary mapping between the indices and MDPs contained in $\Theta$ would have to be established.
> > > >
> > > > > Note that if the true MDP is not in the support of your prior, it cannot be in the support of the posterior, so the inferred MDP can be more and more dissimilar to the true MDP as the MDP space grows.
> > > >
> > > > In L60, we note that the unknown transition function of the underlying MDP has parameters $\theta \in \Theta$. We recognize the lack of clarity here and will update the verbatim of Assumption 1 to make this realizability assumption more apparent to readers.
> > > >
> > > > > I'm just trying to understand if this setting is actually one that the RL community is interested in. Given the other reviews, it appears that my judgement may have been wrong and that the community is in fact interested in this setting -- but can you point me to any work that investigates a setting with a similar assumption? Or are you arguing that this is a novel setting that RL theory should focus on?
> > > >
> > > > To the best of our knowledge, there is no other work on BAMDPs that makes this particular structural assumption. Most work, in their consideration of tabular MDPs, default to a standard model (sometimes identified as the flat Dirichlet-multinomial or FDM distribution) where individual state-action transition distributions are modeled via Dirichlet priors/posteriors. Moreover, prior work is largely concerned with immediately developing an algorithm for approximate Bayesian reinforcement learning; in contrast, while our work does include some novel algorithms, they are present only in service of elucidating the role of the information horizon and how it accurately portrays the hardness of BAMDP planning. It would be premature for us to claim that this setting will be widely of interest to the community at large as this work is, in a sense, a first of its kind (similar to predicting the impact of a research paper, we certainly aim for such an effect, but it would be disingenuous to guarantee it). Any advocacy of this work  on our part centers on the information horizon and, more broadly, regard for underlying information structure when assessing BAMDPs, rather than any individual structural assumptions that solely exist to aid in our presentation of why such ideas are salient and important.
> > > >
> > > > The concern the reviewer raises regarding this assumption seems to be aligned with Weakness 3 of RbYxQ; we’ll rehash some of our response in a bit more detail here. Namely, one might reasonably index a class of MDPs (in this work, transition models specifically though one could also naturally extend to reward functions as well) by some parameter or context, which could simply be a vector of known dimensionality [1, 2]. Under that setup, a distribution over MDPs amounts to a distribution (for example, a Gaussian) over these contexts.  Of course, one could still realize Assumption 2  or an epistemic state abstraction  via an appropriate covering over distribution parameters. What Assumption 1 mitigates, however, is that a single epistemic state (MDP distribution) is continuous and, therefore, requires assessing its corresponding uncertainty via differential entropy, which lacks various desirable properties of traditional Shannon entropy (L181-185).  As we suggested in response to RbyXQ, one feasible way to address this while also taking account for the resulting approximation error is via rate-distortion theory [3].

---

> > > > > ### Author Response · Authors · 2022-08-04
> > > > > **Reply to Reviewer Gwj8 (2/2)**
> > > > >
> > > > > For any epistemic state $p$ denoting a continuous distribution, we have an associated random variable for the MDP $\theta \sim p$. Since we would like this to instead be  a discrete random variable, we could employ the associated rate-distortion function $$\mathcal{R}(D) = \inf\limits_{Z} \mathbb{I}(\theta;Z) \text{ such that } \mathbb{E}\left[d(\theta,Z)\right] \leq D$$ where $\mathbb{I}(\theta;Z)$ denotes the mutual information between $\theta$ and $Z$, the infimum is taken over all discrete random variables $Z$ taking values on a finite set $\mathcal{Z}$ (or, equivalently, over all stochastic Markov kernels/channels mapping between $\Theta$ and $\mathcal{Z}$), $d: \Theta \times \mathcal{Z} \rightarrow \mathbb{R}_{\geq 0}$
> > > > >
> > > > > is a distortion function that assigns non-negative real values quantifying the loss of fidelity by using a value $z \in \mathcal{Z}$ in lieu of $\theta \in \Theta$, and $D \in \mathbb{R}_{\geq 0}$ is a distortion threshold. In words, the rate-distortion function represents the fundamental limit of lossy compression to within a desired expected error threshold. Such lossy compressions have been well-studied by the information theory community for decades and are well-defined even for random variables defined on abstract alphabets [4, 5]. It is perhaps also worth mentioning that the $\delta$-covers and associated notion of metric entropy are intimately related to rate-distortion theory [6].
> > > > >
> > > > > Returning back to our context, one could now operate with any arbitrary parameterization for epistemic states in the original BAMDP and then consider the rate-distortion problem induced at each epistemic state for characterizing a well-defined discrete distribution in adherence to Assumption 1. We would be happy to provide a discussion of this in the next iteration of the paper alongside the current contents; we worry that replacing the current presentation which more simply leans on Assumption 1 might make our contributions seem somewhat esoteric, distracting from the paper’s objective concerning the information horizon itself.
> > > > >
> > > > > [1] Hallak, Assaf, Dotan Di Castro, and Shie Mannor. "Contextual markov decision processes." arXiv preprint arXiv:1502.02259 (2015).
> > > > >
> > > > > [2] Modi, Aditya, Nan Jiang, Satinder Singh, and Ambuj Tewari. "Markov decision processes with continuous side information." In Algorithmic Learning Theory, pp. 597-618. PMLR, 2018.
> > > > >
> > > > > [3] Cover, Thomas M., and Joy A. Thomas. "Elements of information theory 2nd edition (wiley series in telecommunications and signal processing)." Acessado em (2006).
> > > > >
> > > > > [4] Csiszár, Imre. On an extremum problem of information theory. Studia Scientiarum Mathematicarum Hungarica, 9, 1974.
> > > > >
> > > > > [5] Kostina, Victoria, and Ertem Tuncel. "Successive refinement of abstract sources." IEEE Transactions on Information Theory 65, no. 10 (2019): 6385-6398.
> > > > >
> > > > > [6] Kawabata, Tsutomu, and Amir Dembo. "The rate-distortion dimension of sets and measures." IEEE transactions on information theory 40, no. 5 (1994): 1564-1572.

---

### Official Review · Reviewer_bYxQ · 2022-07-11

**Rating:** 6
**Confidence:** 4
**Soundness:** 4 excellent
**Presentation:** 3 good
**Contribution:** 3 good

**Summary:**

 This paper addresses state abstraction in BAMDPs to find approximately Bayes-optimal solutions at reduced computational expense. The paper first introduces the idea of the information horizon: the number of steps at which the true underlying MDP will be identified for any policy. It suggests a simple algorithm to reduce the complexity of solving the BAMDP if the information horizon is less than the BAMDP horizon. This involves finding the optimal Bayes-adaptive policy up until the information horizon, and then executing the optimal policy in the identified MDP thereafter.

For most realistic examples, we cannot expect the ambiguity over the underlying MDP to be completely removed in a small (or indeed, finite) number of steps. Therefore, we can't necessarily expect the information horizon to be less than the BAMDP horizon. To address this, the paper suggests to use state abstraction over the information states. Depending on the coarsity of the abstraction, more of the original information states will be considered to be "close enough" to having resolved the ambiguity over the MDP. Therefore, for more coarse abstractions, we can expect the information horizon to be reduced in the abstract BAMDP. The paper analyses the sub-optimality induced by considering the abstract version of the BAMDP.

**Questions:**

The informed value iteration idea means that once the information horizon has been reached, we just solve standard MDPs, and only need to find the Bayes-optimal solution up until the information horizon. It seems to me that existing sample-based algorithms already become equivalent to solving standard MDPs after the information horizon is reached.

Lets consider BAMCP [37], or indeed any algorithm which makes use of root sampling (sampling a candidate MDP from the initial state to simulate each rollout). Let's also imagine that the information horizon is 1 (i.e. the ambiguity over the MDP is resolved after one step). To make the example very simple, let’s assume the ambiguity is resolved as follows: if, after the first step the agent reaches states in set A, then the MDP is known to be MDP_A, and if the agent reaches states in set B, then the MDP is known to be MDP_B.

Then, all trials that reach set A after the first step are using MDP_A, and all trials that reach set B after the first step are using MDP_B. This means that after the first step, BAMCP reduces to performing standard MCTS in two standard MDPs (i.e. standard MCTS in MDP_A for states in set A and standard MCTS in MDP_B for states in set B).  The point I am trying to make is that if ambiguity over the MDP has been resolved, algorithms which search forward over reachable information states (including RL algorithms) already have the property that they become equivalent to solving individual MDPs after reaching the information horizon.

If this is true, this limits the impact of the work as I don't think the insight of informed value iteration vs value iteration will translate into improvements into other algorithms (i.e. knowledge of the information horizon can only improve performance for naïve value iteration over the entire BAMDP, but wouldn't improve BAMCP-like algorithms).

Summary of questions to the authors:

* Please address weaknesses 1, 2, 3 above
* Is it fair to say that sampled-based algorithms such as BAMCP already implicitly make use of the information horizon as discussed above?
* Can utilising the information horizon more explicitly as proposed in your work make any improvement in the efficiency of existing sample-based (or RL) algorithms, which simulate rollouts in candidate MDPs rather than iterating over the entire belief space?


**Limitations:**

More discussion of limitations due to finite set of MDPs, and applicability of the idea to improve sampled-based/RL algorithms (rather than only improving the most naïve value iteration approach) would improve the paper.

**Strengths And Weaknesses:**

The paper is well written and easy to follow. I think this idea of considering an appropriate information horizon is both interesting and can potentially be an impactful idea. I think the analysis of state abstraction in BAMDPs can be useful for informing approximate algorithms.

There are several shortcomings that I think the paper could address better:

Weakness 1. The paper uses value iteration over the entire information state space to analyse the complexity of informed value iteration vs value iteration, and to demonstrate how using the information horizon can reduce the complexity of value iteration. However, most algorithms for BAMDPS do not iterate over the entire state space, but only compute solutions over the reachable information state space. To me, it seems like these existing algorithms already implicitly take advantage of the fact that some problems have a reduced information horizon, by becoming equivalent to solving standard MDPs once the information horizon is reached. I think this reduces the impact of the informed value iteration idea, as it seems that other common algorithms already (implicitly) do something similar, and this performance improvement might only be applicable to naïve value iteration (which wouldn't be used in practice for BAMDPs). I will elaborate on this in the questions section.

I still think that the paper is good overall since explicitly reasoning about the information horizon allows for providing theoretical bounds. However, it would benefit from discussing this if this is the case.

Weakness 2. Both the informed VI, and informed abstract VI algorithms require knowing what the information horizon is for the given (abstract) BAMDP. However, it is unclear how the information horizon would be determined for a given problem, even for an abstract BAMDP with a known \delta used to compute the state abstraction. It is also not clear when we can expect the information horizon to be decreased when using the state abstraction technique. I would appreciate a greater discussion of this in the paper and in rebuttal.

Weakness 3. Most BAMDP problems (even for finite state and action spaces) assume that the space of possible MDPs is infinite. However, in Assumption 1, this paper assumes that the uncertainty is over a finite set of MDPs. Therefore, to match Assumption 1, additional discretisation may be required over the space of MDPs to make it finite. The paper does not discuss this issue, and would be strengthened if it did also analyse the error introduced by discretising the space of possible MDPs to match Assumption 1. Is it possible to apply the abstraction if the space of possible MDPs is infinite? Please discuss

Minor comments:

* In the preliminaries, \pi is initially used to indicate a non-stationary deterministic policy. However, when discussing the existence of an optimal policy, \pi is then used to refer to a stationary deterministic policy. Please make this clearer.

* Presentation of Bellman equations needs further formatting (e.g. they overflow for BAMDPs and the use of commas and full stops looks a bit arbitrary)

* When discussing the information horizon, authors say I<=H but immediately after discussing the case where I=\infty. Make it more precise.

---

> ### Author Response · Authors · 2022-08-02
> **Response to Reviewer bYxQ (1/3)**
>
> We thank the reviewer for their detailed assessment of our paper. We appreciate the reviewer’s comments acknowledging the novelty, presentation, and potential impact of our contributions.
>
> > In the preliminaries, \pi is initially used to indicate a non-stationary deterministic policy. However, when discussing the existence of an optimal policy, \pi is then used to refer to a stationary deterministic policy. Please make this clearer.
>
> We thank the reviewer for catching this typo on Section 2.1. The supremum defining the optimal policy should indeed be over the non-stationary policy class $\Pi^H$.
>
> > Presentation of Bellman equations needs further formatting (e.g. they overflow for BAMDPs and the use of commas and full stops looks a bit arbitrary)
>
> Again, we thank the reviewer for pointing this out and will clean this up in the next iteration of the paper.
>
> > When discussing the information horizon, authors say I<=H but immediately after discussing the case where I=\infty. Make it more precise.
>
> We believe the reviewer is directing this comment towards L188-L197. Recall the information horizon of a particular policy is defined by an infimum and the information horizon takes an additional supremum over all such non-stationary policies. If there exists a single non-stationary policy that does not succeed in exhausting epistemic uncertainty within $H$ timesteps, then the infimum of its associated information horizon does not exist and so the overall information horizon is infinite. This is what is meant in L188 by the phrase “when it exists” and the exposition of L193-196 is meant to clarify this.
>
> > Weakness 1. The paper uses value iteration over the entire information state space to analyse the complexity of informed value iteration vs value iteration, and to demonstrate how using the information horizon can reduce the complexity of value iteration. However, most algorithms for BAMDPS do not iterate over the entire state space, but only compute solutions over the reachable information state space. To me, it seems like these existing algorithms already implicitly take advantage of the fact that some problems have a reduced information horizon, by becoming equivalent to solving standard MDPs once the information horizon is reached. I think this reduces the impact of the informed value iteration idea, as it seems that other common algorithms already (implicitly) do something similar, and this performance improvement might only be applicable to naïve value iteration (which wouldn't be used in practice for BAMDPs). I will elaborate on this in the questions section.
>
> > Is it fair to say that sampled-based algorithms such as BAMCP already implicitly make use of the information horizon as discussed above?
>
> We thank the reviewer for this detailed question along with a clarifying BAMCP example. Notice that the discussion of the information horizon in L171-L187 is done with respect to BAMDP policies, rather than MDP policies; that is, a policy mapping a hyperstate to an action, rather than only mapping a MDP state. We apologize for not including notation adjacent to $\pi$ in L171 to clarify this (we’ll gladly add this in the next version of the paper).
>
> Sampling-based approaches like BAMCP in large part motivate our work as they represent an attempt to avoid dealing with the overall BAMDP by drawing a single, statistically-plausible transition function and then collapsing down to dealing with the resulting MDP. Consequently, once this downgrading from BAMDP to sampled MDP has occurred, there is no longer a notion of information horizon (as epistemic uncertainty has not actually been resolved but, instead, has been momentarily cast aside and the sampled candidate MDP is being treated as reality). Per Section 3.4 of the BAMCP paper, recall that the rollout policy is the $\epsilon$-greedy policy associated with the optimal $Q$-function of the sampled MDP. In other words, rather than being a BAMDP algorithm that strives to reach the information horizon, posterior-sampling algorithms like BAMCP are built on a heuristic of assuming the underlying information horizon is 1 such that any MDP rollout policy is effectively a viable BAMDP policy operating when epistemic uncertainty has been reduced to zero.
>
> In contrast, the goal of this work is to re-examine the extent to which such deviations from the original BAMDP problem formulation are warranted or even necessary in some cases by introducing the information horizon as a corresponding hardness measure.

---

> > ### Author Response · Authors · 2022-08-02
> > **Response to Reviewer bYxQ (2/3)**
> >
> > > Weakness 2. Both the informed VI, and informed abstract VI algorithms require knowing what the information horizon is for the given (abstract) BAMDP. However, it is unclear how the information horizon would be determined for a given problem, even for an abstract BAMDP with a known \delta used to compute the state abstraction. It is also not clear when we can expect the information horizon to be decreased when using the state abstraction technique. I would appreciate a greater discussion of this in the paper and in rebuttal.
> >
> > We thank the reviewer for raising this point. Not unlike some other theoretical pieces of work [1-3], our paper isn’t meant to advocate for the empirical use of the information horizon as much as it is meant to emphasize it as a tool for analysis. The structure of the planning algorithms introduced in this work clarify exactly how the information horizon might fulfill such a role. We anticipate that practical algorithms for solving BAMDPs will take inspiration from paying attention to information structures in the same manner as our VI-based analysis. If the reviewer has not done so already, we kindly invite them to visit Section C of the appendix which aims to provide discussion on the relationship between epistemic state abstraction and the information horizon. Aside from the condition discussed in Section C.2, another key consideration that requires further investigation is whether an abstract hyperstate containing one of the vertices of the simplex (a hyperstate that is meant to denote a resolution of epistemic uncertainty) can be left once entered. If an agent cannot leave the abstract hyperstates corresponding to such vertices, then a state abstraction of reasonable resolution should be able to incur a reduced information horizon through a hitting-time argument. What remains to be understood is precisely the kinds of paths an agent might take through the epistemic state space and what those trajectories imply at the level of encountered abstract epistemic states.
> >
> > > Weakness 3. Most BAMDP problems (even for finite state and action spaces) assume that the space of possible MDPs is infinite. However, in Assumption 1, this paper assumes that the uncertainty is over a finite set of MDPs. Therefore, to match Assumption 1, additional discretisation may be required over the space of MDPs to make it finite. The paper does not discuss this issue, and would be strengthened if it did also analyse the error introduced by discretising the space of possible MDPs to match Assumption 1. Is it possible to apply the abstraction if the space of possible MDPs is infinite? Please discuss
> >
> > The reviewer is correct that our analysis implicitly makes a sort of realizability assumption such that all value functions are accurately represented under Assumption 1. Notice, however, that accounting for Assumption 1 is not unlike keeping track of approximation error in supervised learning; namely, the finite set of MDPs under consideration constitute a sort of hypothesis class and the best hypothesis (the true MDP transition function) may reside outside this class. Thus, accounting for this error in the discretization amounts to running the analysis while appropriately projecting (external) transition functions onto this class. Our impression is that the textual space and mathematical machinery needed to formalize this would detract from our more central message and presentation around the impact of the information horizon while only providing marginal utility to the theoretical results as they are already presented.
> >
> > Fundamentally, the state abstraction problem is a form of lossy compression problem [4] where, in this context, our aim is to identify a simpler representation of an overwhelmingly large epistemic state space. The use of Assumption 1 in this work is to help establish a proof of concept for such an approach. Still, the reviewer raises a good point that the paper ought to suggest some concrete path for removing this assumption. To that end, we will include a pointer to rate-distortion theory [5] which provides formal tools for dealing with such lossy compression problems regardless of whether random variables are discrete or continuous. We strongly suspect that by considering parameterized classes of MDPs (as done in contextual RL [6]), conditions like Assumption 1 might be replaced by, for example, Lipschitz continuity [7].

---

> > > ### Author Response · Authors · 2022-08-02
> > > **Response to Reviewer bYxQ (3/3)**
> > >
> > > > Can utilising the information horizon more explicitly as proposed in your work make any improvement in the efficiency of existing sample-based (or RL) algorithms, which simulate rollouts in candidate MDPs rather than iterating over the entire belief space?
> > >
> > > As mentioned in our discussion of Weakness 2, we see the information horizon as a valuable tool for analysis, rather than a catalyst for the design of new algorithms that specifically consume the information horizon as an input. In parallel, we do suspect that our work can inspire new algorithms that leverage the underlying information structure of BAMDPs in order to be amenable to theoretical analysis via the information horizon; this motivation is not unlike that of the Bellman rank [1]. That said, and as mentioned in our discussion of Weakness 1, it is likely that the manipulation or heuristic use of the information horizon is already at play in sample-based algorithms, like BAMCP. We also invite the reviewer to consult our response to RYjRz whose questions led us to an interesting possibility for the design of a provably efficient agent via gradually increasing the information horizon.
> > >
> > >
> > > [1] Jiang, Nan, Akshay Krishnamurthy, Alekh Agarwal, John Langford, and Robert E. Schapire. "Contextual decision processes with low bellman rank are pac-learnable." In International Conference on Machine Learning, pp. 1704-1713. PMLR, 2017.
> > >
> > > [2] Dann, Christoph, Nan Jiang, Akshay Krishnamurthy, Alekh Agarwal, John Langford, and Robert E. Schapire. "On oracle-efficient pac rl with rich observations." Advances in neural information processing systems 31 (2018).
> > >
> > > [3] Dong, Shi, Benjamin Van Roy, and Zhengyuan Zhou. "Simple Agent, Complex Environment: Efficient Reinforcement Learning with Agent States." arXiv preprint arXiv:2102.05261 (2021).
> > >
> > > [4] Abel, David, Dilip Arumugam, Lucas Lehnert, and Michael Littman. "State abstractions for lifelong reinforcement learning." In International Conference on Machine Learning, pp. 10-19. PMLR, 2018.
> > >
> > > [5] Cover, Thomas M., and Joy A. Thomas. "Elements of information theory 2nd edition (wiley series in telecommunications and signal processing)." Acessado em (2006).
> > >
> > > [6] Hallak, Assaf, Dotan Di Castro, and Shie Mannor. "Contextual markov decision processes." arXiv preprint arXiv:1502.02259 (2015).
> > >
> > > [7] Modi, Aditya, Nan Jiang, Satinder Singh, and Ambuj Tewari. "Markov decision processes with continuous side information." In Algorithmic Learning Theory, pp. 597-618. PMLR, 2018.

---

> > ### Comment · Reviewer_bYxQ · 2022-08-04
> > **BAMCP**
> >
> > I would like my point about BAMCP to be clarified. What I meant was that BAMCP actually implicitly considers the information horizon: if the belief collapses into a single MDP, from then on only that MDP will be sampled so the tree search will just be planning for an MDP from then on.
> >
> > In other words, in cases where there is a node in the tree which can only be reached by a single candidate MDP, trials passing through that node will always be simulated by the same MDP - and now the statistics for the nodes thereafter are updated as if only solving a single MDP. To me this seems equivalent to the MDP of having reached the information horizon. Is that the case?

---

> > > ### Author Response · Authors · 2022-08-04
> > > **Reply to Reviewer bYxQ**
> > >
> > > We thank the reviewer for their clarifying question.
> > >
> > > > In other words, in cases where there is a node in the tree which can only be reached by a single candidate MDP, trials passing through that node will always be simulated by the same MDP - and now the statistics for the nodes thereafter are updated as if only solving a single MDP. To me this seems equivalent to the MDP of having reached the information horizon. Is that the case?
> > >
> > > Yes, we believe the reviewer is correct that if, at the root node of the search tree, the current posterior over MDPs is a Dirac delta with support on exactly one MDP, then the algorithmic behavior of BAMCP is identical to having reached the information horizon as there is no longer residual epistemic uncertainty in the underlying MDP.

---

> > > > ### Comment · Reviewer_bYxQ · 2022-08-05
> > > > **BAMCP**
> > > >
> > > > The same would hold for a history that is only feasible under a specific MDP, which would make BAMCP boil down to standard MCTS for an MDP  from that node onwards. Thus, one can argue that that part of the search process is actually implicitly considering the information horizon too.
> > > >
> > > > Could you comment on the advantages of reasoning about the information horizon explicitly, as opposed to the implicit way BAMCP considers it?

---

> > > > > ### Author Response · Authors · 2022-08-08
> > > > > **Reply to Reviewer bYxQ**
> > > > >
> > > > > We thank the reviewer for their comments and question.
> > > > >
> > > > > As the nature of our contributions is theoretical, so to is the value of the information horizon in its relation to other BAMDP planning algorithms. That is, future work may very well benefit from characterizing the performance of sample-based planners like BAMCP in terms of the information horizon so as to assess which approximate planners might best be leveraging underlying information structure most efficiently.
> > > > >
> > > > > Of course, doing this and understanding the interplay between the information horizon and existing algorithms serves as a natural stepping stone towards the design of novel BAMDP planning and/or learning algorithms that are constructed to facilitate an efficient analysis through the information horizon, not unlike the existing line of prior work surrounding the Bellman rank in MDPs.

---

> > > > > > ### Comment · Reviewer_bYxQ · 2022-08-09
> > > > > > **thanks**
> > > > > >
> > > > > > thanks for the clarification

---

### Official Review · Reviewer_e61u · 2022-07-11

**Rating:** 7
**Confidence:** 3
**Soundness:** 3 good
**Presentation:** 4 excellent
**Contribution:** 3 good

**Summary:**

The paper deals with Bayes-adaptive Markov decision processes. The paper presents a complexity measure that quantifies the worst-case difficulty for solving an BAMDPs. The measure, the information horizon, is the longest horizon required for an agent to completely reduce it is uncertainty about either the dynamics or reward of the underlying MDP. This measure can then be used to perform planning in a BAMDP over a reduced horizon, before switching to planning in a fully-known MDP, thus reducing time complexity.

Despite the reduction of time complexity by reducing the planning horizon, time complexity of value iteration in BAMDP is still dominated by the size of the state space. The second contribution of the paper revolves around applying principles of state-abstraction on the hyperstate-space of BAMDPs (state X info-state). This leads to a complexity that depends on the state size of the abstract BAMDP.



**Questions:**

N.A.

**Limitations:**

* The paper discuss the limitations an addresses them. Nonetheless, I wish the authors would find a way to integrate parts of the discussion that appear in the appendix into the main text.

**Strengths And Weaknesses:**

The paper is very well-written. Assumptions are clearly stated and background work is discussed in detail. The quality of presentation is excellent. There is a clear thread and argument structure to follow.

The contribution is solid. BAMDPs are computationally very expensive and this paper proposes, what appears to me, as well-grounded improvements. The impact might be somewhat limited due to the fact that it considers only discrete state-action spaces, but anything else would explode the scope.

I am not an expert in the domain of BAMDPs, but the concepts presented in the paper appear novel, despite the fact that they are closely related to a large body of work, that the authors cite.

---

> ### Author Response · Authors · 2022-08-02
> **Response to Reviewer e61u**
>
> We thank the reviewer for their assessment of our paper. We appreciate the reviewer’s
> comments acknowledging the novelty and compelling nature of our contributions. Using the tenth page available for camera-ready papers, we would certainly work to pull discussion content from the appendix into the main body of the paper.

---

### Official Review · Reviewer_YjRz · 2022-07-12

**Rating:** 7
**Confidence:** 4
**Soundness:** 4 excellent
**Presentation:** 4 excellent
**Contribution:** 2 fair

**Summary:**

This paper considers how to tractably plan in BAMDPs, which are the belief-level MDPs induced by introducing uncertainty over the transition dynamics into a traditional finite-horizon MDP. The paper makes two contributions:

1. The paper coins a notion of "information horizon", which is the first timestep at which the agent is guaranteed to be certain about the environment dynamics under a given policy (the information horizon of a BAMDP itself is the maximum of such horizons over all policies). This horizon makes it possible to decompose the complexity of BAMDP planning into a deterministic planning component after the information horizon, and a (much more expensive) belief-space planning component before the information horizon.
2. The paper also suggests an approximate planning scheme for BAMDPs which projects belief states onto a $\delta$-cover for the space of all beliefs. This cover is constructed to contain all the standard basis vectors of the space of beliefs, which correspond to distributions in which the agent is certain of the dynamics. Consequently, as $\delta$ increases, the agent becomes more likely to project its uncertain beliefs onto a deterministic/certain belief. This decreases the effective information horizon. Theoretical analysis provides bounds for the value of a greedy policy under such an approximate belief for a given $\delta$.

**Questions:**

- Do the authors foresee a particular "killer app" for this technique for which the information horizon is both small and easy to compute? I can imagine some artificial problems for which this is true, but I'm wondering whether there's a general class for which it holds.
- How does the complexity of computing the information horizon compare to the overall complexity of BAMDP planning? Is there any worst-case win obtained by using Algorithm 2 in situations where $\mathcal I$ is not known a priori?

**Limitations:**

I felt the paper was clear in its assumptions and results (such as acknowledging that Assumption 2 will not hold in practice). I do not see obvious negative social impacts specific to this work.

**Strengths And Weaknesses:**

Strengths:

- Crisp exposition throughout. POMDPs and Bayesian RL are not my area, but I felt that the proposed concepts built naturally on top of one another.
- Expansive related work (although given that this is not my area, so I cannot say for sure that it is not overlooking work).
- The value approximation scheme requires fairly weak assumptions on the structure of the MDP, and its utility (relative to precise VI on the BAMDP) is precisely quantified in Section 5.3.

Weaknesses:

- The definition of the information horizon is quite strict. If there is any path taken by a policy $\pi$ under which the agent might still have uncertainty at a given time step $h$, then the information horizon  for $\pi$ must be greater than $h$. This is even worse at the level of the BAMDP, where we take a maximum over all $\pi$, including $\pi$ that do not gather much information because they do not act strategically. Consequently, I'm not sure how many "real" BAMDPs are likely to have a small information horizon.
- Although Algorithm 2 makes fairly weak assumptions, it still requires computations that are likely very expensive in practice, like computation of $\mathcal I$ and standard VI repeated over every possible transition function. Both of these are very expensive operations, and I'm not sure how one would easily compute the information horizon or prune the computation of optimal policies with respect to each $\theta$ in order to make the process more tractable.

Admittedly these weaknesses ("the assumptions are too strong/don't scale") are common to many theory-oriented papers. My accept recommendation below is because I feel this paper has adequately demonstrated that the "information horizon" (and abstractions that decrease effective information horizon) is an interesting concept worthy of further fleshing out, even if the practical questions (like how to make Algorithm 2 more efficient) are not solved in this paper.

---

> ### Author Response · Authors · 2022-08-02
> **Response to Reviewer YjRz**
>
> We appreciate the reviewer’s comments on our paper despite Bayesian reinforcement learning being outside their primary area of expertise.
>
> > The definition of the information horizon is quite strict. If there is any path taken by a policy $\pi$ under which the agent might still have uncertainty at a given time step $h$, then the information horizon for $\pi$ must be greater than $h$. This is even worse at the level of the BAMDP, where we take a maximum over all $\pi$, including $\pi$ that do not gather much information because they do not act strategically. Consequently, I'm not sure how many "real" BAMDPs are likely to have a small information horizon.
>
> > How does the complexity of computing the information horizon compare to the overall complexity of BAMDP planning?
>
> The reviewer is correct in their interpretation of the strictness of the information horizon definition. We call attention to the fact that consideration of all policies when looking at the information horizon in this work is tethered to our choice of a VI-based analysis; our paper serves as an initial proof of concept for characterizing why a notion like information horizon might be impactful in BAMDPs. In reality, however, the reviewer’s comment touches upon the idea that it may be overkill to consider all non-stationary policies when assessing the hardness of a BAMDP. We would gladly leave a remark in the next iteration of the paper mentioning how the judicious selection of smaller or regularized policy classes might yield more actionable notions of BAMDP complexity that apply to “real” problems of interest. Doing so would also help to alleviate the reviewer’s concerns around the complexity of computing the information horizon, which becomes simpler upon the imposition of a restricted policy class and could perhaps be further simplified via other structural assumptions on the class.
>
> Orthogonally and as we note in the reply to RbYxQ, existing algorithms like BAMCP can be interpreted as making heuristic assumptions on the underlying information horizon in order to bypass dealing with the full BAMDP directly. When it comes to learning (as opposed to planning) in BAMDPs, we suspect that an analysis might proceed to examine each policy deployed at each time period and consider their individual information horizons (rather than needing to take a further supremum across the policy class to accommodate a planner like VI).
>
> > Although Algorithm 2 makes fairly weak assumptions, it still requires computations that are likely very expensive in practice, like computation of and standard VI repeated over every possible transition function. Both of these are very expensive operations, and I'm not sure how one would easily compute the information horizon or prune the computation of optimal policies with respect to each in order to make the process more tractable.
>
> Do the authors foresee a particular "killer app" for this technique for which the information horizon is both small and easy to compute? I can imagine some artificial problems for which this is true, but I'm wondering whether there's a general class for which it holds.
>
>
> We thank the reviewer for raising this point. Not unlike some other theoretical pieces of work [1-3], our paper isn’t meant to advocate for the immediate empirical use of the information horizon as much as it is meant to emphasize it as a tool for analysis. The structure of the planning algorithms introduced in this work clarify exactly how the information horizon might fulfill such a role. We anticipate that practical algorithms for solving BAMDPs will take inspiration from paying attention to information structures in a careful manner that would then facilitate a theoretical analysis via the information horizon; this motivation is not unlike that of the Bellman rank [1].
>
> > Is there any worst-case win obtained by using Algorithm 2 in situations where $\mathcal{I}$ is not known a priori?
>
> This is an interesting idea. One possibility we have not yet thought through carefully is to design a BAMDP agent that hypothesizes a gradually increasing information horizon; in [3] the analogue of this idea manifests in the form of an incrementally increasing discount factor, slowly expanding the horizon over which an agent must competently behave in the environment. As noted before, sampling based algorithms like BAMCP that immediately restrict focus to a single sampled MDP are, in effect, acting with an information horizon of 1. Perhaps there is something to be gained by considering this kind of incremental approach where an agent slowly builds up to the complexity of the full BAMDP via an increasing sequence of information horizons. How to best exploit each successive information horizon at each time period is an interesting avenue for future work. We’ll gladly include a short paragraph on this in the next version of the paper

---

> > ### Author Response · Authors · 2022-08-02
> > **References**
> >
> > [1] Jiang, Nan, Akshay Krishnamurthy, Alekh Agarwal, John Langford, and Robert E. Schapire. "Contextual decision processes with low bellman rank are pac-learnable." In International Conference on Machine Learning, pp. 1704-1713. PMLR, 2017.
> >
> > [2] Dann, Christoph, Nan Jiang, Akshay Krishnamurthy, Alekh Agarwal, John Langford, and Robert E. Schapire. "On oracle-efficient pac rl with rich observations." Advances in neural information processing systems 31 (2018).
> >
> > [3] Dong, Shi, Benjamin Van Roy, and Zhengyuan Zhou. "Simple Agent, Complex Environment: Efficient Reinforcement Learning with Agent States." arXiv preprint arXiv:2102.05261 (2021).

---

> > > ### Comment · Reviewer_YjRz · 2022-08-09
> > > **Response to rebuttal**
> > >
> > > Thank you for your comments responding to my original questions and clarifying what you see as the value of this paper. Leaving a comment about shrinking the policy class over which the BAMDP supremum is defined and mentioning the receding information horizon idea both seem like nice things to have (but not mandatory—don't sweat it if you're at the page limit). After reading the other reviews, I'm still in favor of accepting this paper, although I agree with bYxQ's comments suggesting that this complexity measure may have limited algorithmic impact given that existing algorithms seem to be able to exploit problems with short information horizons already.

---

### Meta-Review · Area_Chair_F1VL · 2022-08-27

**Recommendation:** Accept
**Confidence:** Certain

**Metareview:**

In post-rebuttal discussion, reviewers debated the merits of the paper and especially reviewer Gwj8's concerns.  In the end, reviewer Gwj8 agreed with other reviewers that the paper should be accepted, although all reviewers would like to see the final revision reflect the points and concerns raised in the post-rebuttal discussion.


**Award:**

No

---

### Decision · Program_Chairs · 2022-09-14

Accept